# `FedFit`: Federated Dynamic Sparse Training via Fisher Information scoring

**Meng Bi**[1]  **Hong Huang**[2]  **Jinlong Song**[2]  **Charles Wang**[1]  **Chengming Hu**[3]  **Xi Chen**[3]  **Ting Yu**[1]  **Xue Liu**[1 3]

## Abstract

Cross-device Federated Learning (FL) is frequently bottlenecked by the prohibitive memory and communication costs of training deep neural networks on resource-constrained edge hardware. While federated dynamic sparse training aims to alleviate these costs by adjusting sparse structures during training, existing methods rely on magnitude-based heuristics that are fundamentally ill-suited for the non-convergent, heterogeneous environments inherent to FL. To address this challenge, we propose `FedFit`, a federated dynamic sparse training framework that replaces simple heuristics with optimization-centric criteria for structure adjustment. By leveraging a second-order approximation of the loss landscape via the Fisher Information Matrix, `FedFit` enables precise and efficient structure adjustment without the overhead of explicit Hessian computation. Empirical evaluations across computer vision and natural language processing benchmarks demonstrate that `FedFit` significantly narrows the sparse-to-dense accuracy gap, outperforming state-of-the-art methods while maintaining high communication efficiency. Our code is available at https://github.com/Serena-28/Fedfit.git.

## 1. Introduction

Cross-device Federated Learning (FL) (McMahan et al., 2017; Li et al., 2019; Kairouz et al., 2021; Yang et al., 2023; Yang & Zhang, 2021) enables multiple edge devices to collaboratively train deep learning models while preserving

---
[1]Department of Computer Science, Mohamed bin Zayed University of Artificial Intelligence, Abu Dhabi, United Arab Emirates [2]Department of Computer Science, City University of Hong Kong, Hong Kong, China [3]School of Computer Science, McGill University, Montreal, Quebec, Canada. Correspondence to: Hong Huang <honghuang2000@outlook.com>.

*Proceedings of the 43rd International Conference on Machine Learning*, Seoul, South Korea. PMLR 306, 2026. Copyright 2026 by the author(s).

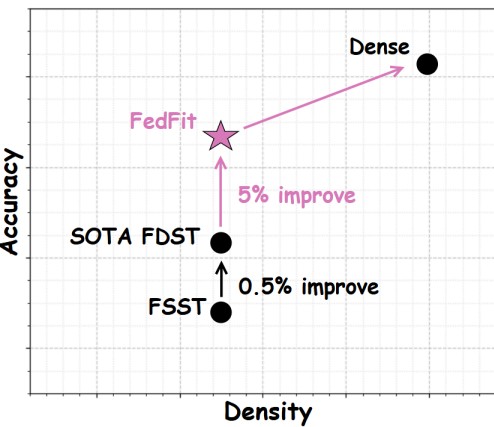

*Figure 1.* Visualizing the value of `FedFit` under a fixed density.

data locality and privacy. Despite its potential, the deployment of deep neural networks in cross-device settings is often bottlenecked by the stringent computational and communication constraints of edge hardware. While traditional neural network pruning (Han et al., 2015; Molchanov et al., 2019; Ma et al., 2021) effectively reduces model complexity, most post-training paradigms are incompatible with FL, as they necessitate an initial resource-intensive dense training phase that violates the very resource constraints FL aims to respect.

To bypass the requirement for dense training, contemporary federated pruning frameworks (Bibikar et al., 2022; Qiu et al., 2022; Tian et al., 2024; Jiang et al., 2022; Huang et al., 2024; 2023; Munir et al., 2021) have integrated techniques from Dynamic Sparse Training (DST) (Evci et al., 2020; Raihan & Aamodt, 2020; Jayakumar et al., 2020). These methods embed model structure adjustments, iterative pruning and reactivation, directly into the FL lifecycle. Typically, these frameworks operate via a nested optimization structure: an inner loop performs standard federated optimization (*e.g.,* FedAvg (McMahan et al., 2017)) on a fixed sparse mask, while a periodic outer loop adjusts the mask structure based on heuristic criteria, as shown in Fig. 2. By co-adapting weights and structure, these methods aim to converge directly onto a specialized sparse model, significantly reducing the cumulative overhead.

However, we observe a critical limitation in current federated pruning frameworks: **sparse structure adjustments**

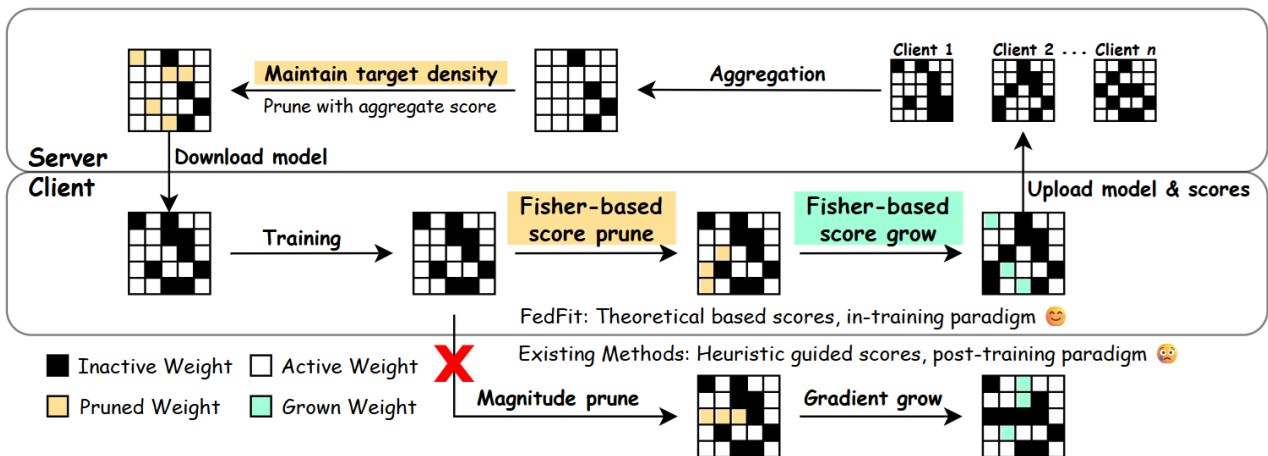

*Figure 2.* Illustration of sparse structure adjustment in FedFit and existing baselines. *Top:* FedFit performs Fisher based score prune and grow on clients, and the server optionally applies global score-based pruning to maintain the target density. *Bottom:* Existing baselines typically rely on heuristic magnitude pruning and gradient based growing, leading to less stable structure updates under heterogeneity.

**often fail to yield reliable performance gains.** Empirical evidence suggests a persistent accuracy gap between models trained via federated dynamic sparse training and their dense counterparts. Moreover, in many cases, performance improvements over baseline strategies, federated static sparse training (FSST) (Xu et al., 2021) remain marginal, as shown in Fig. 1. This suggests a fundamental lack of precision in how these frameworks identify and adapt critical model parameters.

Our analysis reveals that the root cause of this limitation lies in the reliance on **heuristic-driven criteria**, specifically, pruning by weight magnitude and reactivating by gradient magnitude. Through a unified theoretical analysis of the pruning and reactivation, we demonstrate that these heuristics are inherited from **post-training paradigms**. Such paradigms assume that the model is near convergence (where gradients $\approx 0$) and that the Hessian remains locally stable—assumptions that are fundamentally violated in federated dynamic sparse training. In FL, the structure is adjusted while the model is far from convergence, a problem exacerbated by the inherent instability of data heterogeneity. Consequently, these outdated heuristics provide an inaccurate signal for parameter importance, leading to suboptimal sparse structures.

To address these limitations, we propose `FedFit`, a precise federated dynamic sparse training framework that replaces conventional heuristics with **optimization-centric criteria** for structure adjustment. By leveraging both gradient and Hessian information, `FedFit` ensures that the evolution of the model structure is formally aligned with the trajectory of loss minimization. To facilitate practical deployment within resource-constrained federated environments, we approximate the necessary second-order information via the Fisher Information Matrix, providing a high-fidelity proxy

for parameter importance without the prohibitive overhead of explicit Hessian computation. Furthermore, we provide a comprehensive overhead analysis and derive a theoretical upper bound for the approximation error, making `FedFit` not only empirically effective but also theoretically robust in the dynamic training paradigms.

We evaluate `FedFit` on a suite of computer vision (CV) and natural language processing (NLP) benchmarks to demonstrate its effectiveness under diverse federated learning scenarios. Our comprehensive experiments show that `FedFit` consistently achieves higher accuracy and superior generalization compared to state-of-the-art (SOTA) federated pruning methods, while introducing negligible computational and communication overhead. For instance, using a ResNet-18 model in the standard SVHN experiment, `FedFit` achieves a 7.04% accuracy improvement over the best baseline, with 80% the communication saving of the dense model.

Our contributions are summarized as follows:

- We provide, to our knowledge, the first theoretical treatment of the criteria governing federated dynamic sparse training, identifying why traditional heuristics fail under non-convergent, heterogeneous conditions.

- We introduce `FedFit`, a theoretically grounded FDST framework for online prune-and-grow adaptation, using Fisher information as an efficient approximation.

- We evaluate `FedFit` across diverse CV and NLP benchmarks. Results indicate that `FedFit` consistently outperforms state-of-the-art (SOTA) methods in both accuracy and generalization.

## 2. Preliminary and Challenges

### 2.1. Federated Dynamic Sparse Training

Federated Dynamic Sparse Training (FDST) collaboratively optimizes a global sparse model $\theta \odot m$, where $\theta \in \mathbb{R}^M$ represents the model weights and $m \in \{0, 1\}^M$ is a binary mask. Unlike static pruning, FDST allows the mask $m$ to evolve throughout training while maintaining a target density $\rho = \frac{\|m\|_0}{M}$. In each round $t$, clients receive the global state $(\theta_t, m_t)$ with density $\rho$ to initialize their local models $\theta_t^n \odot m_t^n$, and perform sparse training with learning rate $\eta_t$:

$$\theta_{t+1}^n \odot m_t^n = \theta_t^n \odot m_t^n - \eta_t \nabla \mathcal{L}_n(\theta_t^n \odot m_t^n, \mathcal{D}_n) \odot m_t^n, \quad (1)$$

where $\mathcal{L}_n(\cdot)$ is the local loss function with dataset $\mathcal{D}_n$, and $p_n$ is the aggregation weight for client $n$. The core of FDST lies in the **sparse structure adjustment process**. In local-driven adjustment methods (Bibikar et al., 2022; Tian et al., 2024), clients adjust their sparse structure before communication by pruning a fraction of low-importance weights and reactivating an equal number of parameters in previously pruned locations:

$$m_{t+1}^n = \text{Adjust}(m_t^n, \mathcal{P}_n, \mathcal{G}_n), \quad (2)$$

where $\mathcal{P}_n$ and $\mathcal{G}_n$ represent the pruning and reactivation criteria, typically based on weight magnitudes $\mathcal{P}_n = |\theta_{t+1}^n|$ and local gradient magnitudes $\mathcal{G}_n = |\nabla \mathcal{L}_n|$, respectively. Following this, the server aggregates the local weights $\theta_{t+1} = \sum p_n \theta_{t+1}^n$ and performs a union operation $m_{t+1} = \bigcup m_{t+1}^n$, often followed by a global pruning step $m_{t+1} := \text{Prune}(m_{t+1}, |\theta_{t+1}|)$ to restore the target density $\rho$. Alternatively, some server-driven adjustment methods (Huang et al., 2023; 2024) shift the adjustment process to the server to utilize global information, though this typically incurs significantly higher communication overhead due to the transmission of dense gradient information.

By dynamically exploring the parameter space, FDST discovers optimal subnetworks while minimizing on-device memory and computational costs.

### 2.2. Challenges in Federated Dynamic Sparse Training

While FDST is intuitively motivated by the need to discover optimal subnetworks under resource constraints, existing methods face a critical bottleneck: **the sparse structure adjustment process often fails to yield reliable performance gains.** As illustrated by the empirical results in Fig. 2, the performance of state-of-the-art FDST methods does not consistently outperform static sparse training.

This performance plateau suggests a fundamental limitation in current FDST paradigms. Given a limited exploration budget (i.e., a finite number of communication rounds and local epochs), existing structure adjustment methods fail to effectively identify and preserve critical model parameters. This issue is exacerbated at low densities, where a large performance gap remains between FDST and its dense counterpart. Consequently, the core challenge lies in developing more robust adjustment mechanisms that can accurately capture the global importance of parameters despite the data heterogeneity and restricted information exchange inherent in federated learning.

## 3. Methodology

In this section, we propose `FedFit`, motivated by a key limitation of existing federated dynamic sparse training methods: their prune and grow decisions are largely driven by heuristics, which refers to magnitude or gradient based rules, and can be biased and unstable under client heterogeneity. Instead of relying on this, we show what prune and grow should depend on by deriving an objective consistent importance score from a second-order approximation of the loss, yielding a Fisher informed scoring rule for topology adjustment. Building on this score, we present the `FedFit` and its federated training protocol. Finally, we establish a theoretical upper bound for `FedFit`.

### 3.1. Probelm reformulation

FDST aims to jointly optimize model weights and sparse topological structure under communication constraints and non-independent and identically distributed (non-IID) client data. However, we find that the importance signals used to make prune and grow decisions are insufficiently accurate.

Our central claim is that existing FDST frameworks predominantly rely on **heuristic** pruning by weight magnitude and reactivation by gradient magnitude, and these criteria are inherited from the **post-training paradigm**. We will provide the proof in Sec. 3.2. The post-training paradigm typically assumes that (i) the model is close to a stationary point, so the gradient is approximately zero in expectation $g \approx 0$; (ii) The local curvature is relatively stable, with a fixed Hessian matrix $H_{kk}$ (see discussion in Sec.3.2).

However, in FDST structural adjustments occur long before convergence, and data heterogeneity induces substantial drift in both gradients and curvature across clients and rounds. Consequently, importance estimates based on magnitude or single round gradients often exhibit large bias and high variance, ultimately distorting importance assessment and undermining accurate model training.

### 3.2. Fisher-based prune score

Based on the above limitation, we derive the scoring criterion that should guide pruning and growing decisions. Given a neural network, we consider pruning a single parameter $\theta$.

**Assumption 1.** *A small perturbation d is applied to the parameter θ, thus the second-order Taylor expansion provides an adequate truncation. For a sufficiently small d, the higher order term $\sigma(||d||^2)$ is negligible.*

For notational simplicity, we denote $\mathcal{L}_n$ as $\mathcal{L}$ from now on. The change in the loss function $\mathcal{L}(\theta)$, from perturbation $d$ approximated by the second order Taylor expansion is

$$\mathcal{L}(\theta + d) - \mathcal{L}(\theta) \approx g^\top d + \frac{1}{2} d^\top H d. \qquad (3)$$

where $g = \nabla \mathcal{L}(\theta)$ denotes the gradient and $H = \nabla^2 \mathcal{L}(\theta)$ denotes the Hessian matrix.

Pruning the k-th parameter $\theta_k$ corresponds to setting $d = -\theta_k e_k$, where $e_k$ is the $k$-th standard basis vector. This yields the loss increment induced by pruning:

$$\mathcal{L}(\theta - \theta_k e_k) - \mathcal{L}(\theta) \approx -g_k \theta_k + \frac{1}{2} H_{kk} \theta_k^2. \qquad (4)$$

As in Sec.3.1, many existing federated pruning frameworks derived from post-pruning methods assume $g \approx 0$ in expectation over the dataset and ignore the first-order term. Meanwhile, they further approximate $H_{kk}$ as constant, and the result of Eq. (4) becomes

$$\mathcal{L}(\theta - \theta_k e_k) - \mathcal{L}(\theta) \propto |\theta_k|, \qquad (5)$$

and consequently adopt a magnitude-based pruning strategy. However, prune and grow are both dynamic rather than performed only after the model is well trained, so magnitude-based pruning performance degrades in this setting.

Thus, we decide to apply the full criteria in Eq.4. However, in practical settings such as real-world training and resource-constrained federated learning, explicitly constructing or inverting the Hessian is often prohibitively expensive and incurs substantial communication overhead. Therefore, we introduce the following assumption to approximate the second-order curvature in federated learning.

**Assumption 2.** *The Hessian of the loss function is approximated by the Fisher information matrix, $H \approx F$.*

From this assumption, we use the Fisher information matrix in place of the Hessian to characterize parameter sensitivity and importance, avoiding explicit second-order computation. The theoretical proof is provided in Sec B.1. With Assumption 2, the Fisher based pruning score is defined as:

$$\mathcal{P}_k = -g_k \theta_k + \frac{1}{2} \theta_k^2 F_{kk}. \qquad (6)$$

We prioritize pruning the parameters with smaller $P_n$.

Since full Fisher $F \in \mathbb{R}^{d \times d}$ is large, we adopt the Kronecker-factored approximate curvature (K-FAC)

method (Grosse & Martens, 2016) to approximate $F \in \mathbb{R}^{d \times d}$ under resource constrained networks. Concretely, for layer $\ell$, K-FAC approximates the corresponding Fisher block as $F_\ell \approx A_\ell \otimes G_\ell$, where $A_\ell$ denotes the second moment of the layer input activations, and $G_\ell$ denotes the second moment of the backpropagated pre-activation gradients, with implementation details deferred to Sec B.2.

### 3.3. Fisher-based grow score

For the growing strategy, we reformulate the grow problem as estimating the maximum decrease in the loss $\mathcal{L}$ after reactivating a parameter. We use this quantity as the grow signal, and use the same setting as in pruning, letting $\theta$ denote the model parameters.

In one grow operation, we allow only the $j$-th parameter to change, keeping all other coordinates fixed. We take the increment $d = \alpha e_j$, where $e_j$ is the $j$-th standard basis vector, and $\alpha$ is the growth step size. A second-order Taylor expansion of $\mathcal{L}(\theta + \alpha e_j)$ around $\theta$ along $e_j$ yields

$$\mathcal{L}(\theta + \alpha e_j) \approx \mathcal{L}(\theta) + g_j \alpha + \frac{1}{2} H_{jj} \alpha^2. \qquad (7)$$

Applying Assumption 2, we approximate the local curvature with the empirical Fisher. The predicted loss decrease induced by activating the $j$-th parameter is then $q_j(\alpha) = \mathcal{L}(\theta) - \mathcal{L}(\theta + \alpha e_j)$. Since $F_{\text{emp}}$ is positive semidefinite (see proofs in Sec. B.3), its diagonal entries are nonnegative, ensuring $q_j(\alpha)$ is a concave quadratic function of $\alpha$ under the second-order approximation. The optimal step size $\alpha_j^\star$ is then obtained by differentiating $q_j(\alpha)$. Finally, the single-parameter growth score is given by

$$\mathcal{G}_j = \frac{1}{2} \frac{g_j^2}{(F_{\text{emp}})_{jj}}. \qquad (8)$$

We prioritize growing the parameters with larger $\mathcal{G}_j$.

### 3.4. Second-Order Analysis for Layerwise

Consider a fixed layer $\ell$ with parameter vector $\theta_\ell$ and a pruning index set $S_\ell$. Pruning all elements in $S_\ell$ is equivalent to applying a perturbation to the layer parameters, given by $d_\ell = -\theta_\ell \odot \mathbf{1}_{S_\ell}$. Under Assumption 1, a second-order Taylor expansion of the loss yields the predicted loss increment $\Delta \mathcal{L}_\ell(S_\ell) = \mathcal{L}(\theta_\ell + d_\ell) - \mathcal{L}(\theta_\ell)$. Following Assumption 2, we approximate the Hessian $H_\ell$ with the layerwise empirical Fisher information matrix $F_\ell$. This gives

$$\Delta \mathcal{L}_\ell(S_\ell) \approx g_\ell^\top d_\ell + \frac{1}{2} d_\ell^\top F_\ell d_\ell, \qquad (9)$$

where $g_\ell = \nabla_{\theta_\ell} \mathcal{L}(\theta_\ell)$ and $H_\ell = \nabla_{\theta_\ell}^2 \mathcal{L}(\theta_\ell)$. This objective can be naturally decomposed into (i) a sum of the individual contributions of pruned parameters in $S_\ell$ and (ii) an intralayer interaction residual induced by off-diagonal Fisher

entries. Specifically, we can write

$$\Delta\mathcal{L}_\ell(S_\ell) \approx \sum_{i \in S_\ell} \mathcal{P}_i + \mathcal{R}_\ell(S_\ell), \tag{10}$$

where $\mathcal{P}_i$ denotes the loss increment induced by pruning a single parameter and $\mathcal{R}_\ell(S_\ell)$ collects the off-diagonal interaction terms. Directly optimizing Eq.(10) results in a quadratic combinatorial optimization problem and is thus impractical in federated settings. Therefore, we adopt a separable surrogate objective that focuses on the parameter-wise terms and select

$$S_\ell = \arg\min_{|S|=m_\ell} \sum_{i \in S} \mathcal{P}_i. \tag{11}$$

That is, within layer $\ell$, we prune the $m_\ell$ parameters with the smallest $\mathcal{P}_i$, where the pruning budget is $m_\ell = (1 - \rho_\ell)p_\ell$ under the layer-wise target density $\rho_\ell$ determined by the ERK allocation rule (Evci et al., 2020) and $p_\ell$ denotes the number of parameters in layer $\ell$.

We defer the explicit expansion of $\mathcal{R}_\ell(S_\ell)$ and its norm-based upper bound to Appendix B.4.

The growing step follows the same second-order logic as pruning: we select parameters that maximize the predicted loss decrease under a single-coordinate activation. Accordingly, we define a layerwise growing set $G_\ell$ by selecting the $m_\ell$ largest growth scores $\mathcal{G}_i$ within layer $\ell$, where $\mathcal{G}_i$ is the single-parameter growth score defined in Section 3.3,

$$G_\ell = \arg\max_{|G|=m_\ell} \sum_{i \in G} \mathcal{G}_i. \tag{12}$$

In this way, the grow budget matches the prune budget, keeping the ERK-assigned layer-wise density $\rho_\ell$ unchanged. The full layerwise growing derivation is provided in Appendix B.5.

### 3.5. Proposed framework

Based on the above derivations of the pruning and growing scores, we propose `FedFit`, which is a new Fisher information-guided FDST framework, as illustrated in Fig. 2. `FedFit` follows a nested two level (inner-outer) iteration scheme. On the client side, structural adjustment is performed using our prune and grow scores, where the Hessian is replaced by a Fisher K-FAC approximation, enabling more robust topology updates. The updated sparse structure and scores are then uploaded to the server. After aggregation, the server can further apply a global fisher based score pruning step to restore the target density. `FedFit` repeats this process until convergence, and detailed procedures are provided in Algorithm 1 and Section C.1.

By introducing Fisher-guided scores, `FedFit` provides an interpretable and tractable criterion for topology adjustment.

Compared with the heuristic strategies used in prior work, this score-driven adaptive adjustment improves the stability and robustness of the learned sparse structure and yields better final performance in federated settings.

### 3.6. Overhead Analysis

We analyze the computation and communication overhead introduced by `FedFit` during topology adjustment.

**Computation Overhead.** The overhead of `FedFit` arises from two steps on the client: (i) computing prune and grow scores for all parameters and (ii) estimating Fisher information matrix through the K-FAC approximation.

Let $n$ denote the total number of trainable parameters in a client. Computing prune and grow scores for all parameters takes $\mathcal{O}(n)$ time per adjustment. The additional overhead from the K-FAC Fisher approximation is $\mathcal{O}(B \sum_\ell (d_{\text{in},\ell} + d_{\text{out},\ell}))$, as given in appendix B.6. In a prune and grow round, the additional computational overhead is $\mathcal{O}(n + B \sum_\ell (d_{\text{in},\ell} + d_{\text{out},\ell}))$. If pruning and growing is performed once every $R$ communication rounds, the above cost is amortized by a factor of $R$ per round. For common architectures where most layers are approximately square ($d_{\text{in},\ell} \approx d_{\text{out},\ell}$), summing over layers yields $\sum_\ell (d_{\text{in},\ell} + d_{\text{out},\ell}) = \mathcal{O}(\sum_\ell d_{\text{in},\ell} d_{\text{out},\ell}) = \mathcal{O}(n)$. Therefore the additional computation is $\mathcal{O}(n + Bn) = \mathcal{O}(n)$, and is linear in the number of parameters.

**Communication Overhead.** Exchanges between the server and clients occur only after each topology adjustment. At each communication round, `FedFit` client uploads model parameters and Fisher prune scores to the server, which is then aggregated, and the model is sent back to clients. Similarly to FedDST, `FedFit` exchanges only sparse model states throughout training, so the parameter communication cost is $\mathcal{O}(n)$. The additional cost comes from communicating scores, which are transmitted only for the same active sparse coordinates and hence have the same order $\mathcal{O}(n)$. In other words, the upload cost is additive, thus we have $\mathcal{O}(n + n) = \mathcal{O}(n)$. Moreover, all Fisher related statistics are computed locally on clients and do not incur any additional communication. Communication costs are calculated as described in Sec. B.8.

### 3.7. Theoretical Upper Bound

`FedFit` introduces two main sources of error in pruning and growing: (i) second-order Taylor truncation error and (ii) Fisher-information approximation error.

**Assumption 3** (Hessian Lipschitzness)**.** *The loss function $\mathcal{L}(\theta)$ is three times continuously differentiable, and its Hessian is Lipschitz. That is, there exists a constant $\rho_H > 0$ such that for any $\theta$ and perturbation $d$,*

*Table 1.* Test accuracy (%) of ResNet18 and ShuffleNetV2 across multiple datasets under varying target densities $\rho$. The best results are highlighted in green, and the second-best results are marked in yellow. FedAvg is density-independent and thus reports a single value shared across all $\rho$. All results are reported as mean±std.

| | Method | CIFAR10 | | | CINIC10 | | |
| --- | --- | --- | --- | --- | --- | --- | --- |
| | | $\rho = 0.1$ | $\rho = 0.2$ | $\rho = 0.3$ | $\rho = 0.1$ | $\rho = 0.2$ | $\rho = 0.3$ |
| ResNet18 | FedAvg | | 77.25±1.97 | | | 66.76±4.39 | |
| | FSST | 61.57±2.79 | 70.39±1.70 | 75.44±2.56 | 49.30±1.87 | 57.80±4.12 | 62.32±3.08 |
| | FedDST | 63.13±2.53 | 68.31±2.51 | 74.64±2.77 | 53.22±2.64 | 58.42±2.65 | 59.78±1.66 |
| | FedTiny | 63.21±2.99 | 69.38±3.32 | 74.00±2.32 | 52.28±3.10 | 59.48±2.86 | 61.92±2.44 |
| | FedMef | 59.38±5.61 | 71.56±2.38 | 74.04±4.80 | 54.34±2.18 | 58.62±2.43 | 60.36±2.31 |
| | FedSGC | 63.06±2.31 | 74.23±2.37 | 74.23±2.37 | 54.90±1.93 | 60.08±2.78 | 62.17±2.01 |
| | FedRTS | 64.54±2.79 | 73.12±2.41 | 75.08±2.07 | 55.52±4.06 | 61.08±2.29 | 62.15±2.51 |
| | FedFit | **65.51±1.25** | **74.73±1.07** | **76.43±1.72** | **57.94±2.01** | **63.11±2.39** | **65.69±2.39** |
| ShuffleNet | FedAvg | | 60.26±2.87 | | | 53.77±2.01 | |
| | FSST | 38.40±2.07 | 56.33±2.50 | 59.06±3.25 | 42.11±1.68 | 47.50±2.43 | 49.22±2.70 |
| | FedDST | 47.44±2.20 | 57.93±2.65 | 59.59±3.23 | 41.17±6.46 | 47.06±2.81 | 48.45±2.08 |
| | FedTiny | 41.68±3.37 | 57.35±2.48 | 61.02±2.98 | 44.00±1.79 | 48.43±2.44 | 48.90±2.20 |
| | FedMef | 46.09±2.59 | 56.39±2.16 | 59.26±3.25 | 44.52±2.01 | 48.55±2.29 | 49.09±2.15 |
| | FedSGC | 41.16±1.95 | 55.33±2.38 | 59.68±2.99 | 42.48±2.16 | 47.82±2.41 | 49.23±2.09 |
| | FedRTS | 45.98±1.31 | 58.02±2.49 | 60.80±2.88 | 44.18±1.93 | 48.10±2.25 | 49.34±2.91 |
| | FedFit | **50.21±2.31** | **58.57±3.33** | **61.03±3.19** | **44.57±1.94** | **50.69±2.91** | **53.26±2.68** |

| | Method | SVHN | | | Tiny ImageNet | | |
| --- | --- | --- | --- | --- | --- | --- | --- |
| | | $\rho = 0.1$ | $\rho = 0.2$ | $\rho = 0.3$ | $\rho = 0.1$ | $\rho = 0.2$ | $\rho = 0.3$ |
| ResNet18 | FedAvg | | 92.85±1.23 | | | 8.97±0.23 | |
| | FSST | 76.88±2.20 | 82.91±5.62 | 84.25±3.26 | 4.12±0.44 | 4.97±0.22 | 7.02±0.56 |
| | FedDST | 74.73±5.33 | 81.47±4.60 | 81.79±5.77 | 3.30±0.10 | 4.70±0.10 | 5.30±0.08 |
| | FedTiny | 79.38±3.40 | 83.21±3.95 | 83.12±6.03 | 3.35±0.17 | 4.80±0.13 | 5.25±0.12 |
| | FedMef | 80.81±3.71 | 82.82±4.59 | 83.21±5.75 | 3.32±0.12 | 4.72±0.12 | 5.22±0.10 |
| | FedSGC | 81.58±3.90 | 83.46±5.50 | 83.74±4.38 | 4.21±0.20 | 5.06±0.22 | 7.29±0.18 |
| | FedRTS | 81.25±3.77 | 82.44±3.96 | 83.83±4.63 | 4.25±0.17 | **5.60±0.21** | 7.42±0.18 |
| | FedFit | **86.14±3.04** | **90.25±1.23** | **91.26±1.37** | **4.99±0.72** | 5.38±0.41 | **7.61±0.52** |
| ShuffleNet | FedAvg | | 69.23±5.16 | | | 5.87±0.45 | |
| | FSST | 13.80±3.18 | 60.43±6.08 | 70.45±4.30 | 3.18±0.53 | 4.56±0.35 | 5.44±0.47 |
| | FedDST | 36.20±3.99 | 63.62±6.55 | 68.98±5.91 | 3.30±0.10 | 4.70±0.10 | 5.30±0.08 |
| | FedTiny | 14.82±3.24 | 64.18±6.07 | 67.03±6.26 | 3.35±0.17 | 4.80±0.13 | 5.25±0.12 |
| | FedMef | 20.52±3.16 | 63.52±6.07 | 67.56±5.72 | 3.32±0.12 | 4.72±0.12 | 5.22±0.10 |
| | FedSGC | 13.99±3.46 | 58.57±4.80 | 70.24±5.56 | 3.25±0.01 | 4.58±0.10 | 5.15±0.12 |
| | FedRTS | 33.67±3.72 | 63.71±6.17 | 70.25±5.07 | 3.70±0.20 | 4.90±0.12 | 5.40±0.11 |
| | FedFit | **50.63±4.73** | **67.09±6.26** | 69.15±5.94 | **4.09±0.13** | **5.15±0.30** | **5.63±0.49** |

$\|H(\theta + d) - H(\theta)\| \leq \rho_H \|d\|_2$, *where* $H(\theta) = \nabla^2 \mathcal{L}(\theta)$ *and* $\| \cdot \|$ *denotes the spectral norm.*

**Assumption 4** (Fisher approximation quality). *Let* $F(\theta)$ *be the exact Fisher information matrix, and let* $\widehat{F}(\theta)$ *be the K-FAC approximation used in* FedFit*. There exists a constant* $\delta_F(\theta) \geq 0$ *such that* $\|F(\theta) - \widehat{F}(\theta)\| \leq \delta_F(\theta)$.

Under Assumptions 3 and 4, we can decompose the second-order score error into two terms and obtain upper bounds.

**Second-order Taylor truncation error.** Around a reference point $\theta$, FedFit applies a second-order expansion to the parameter perturbation $d$ induced by topology adjustment:

$$\mathcal{L}(\theta + d) = \mathcal{L}(\theta) + \langle \nabla \mathcal{L}(\theta), d \rangle \\ + \frac{1}{2} d^\top H(\theta) d + \mathcal{R}(\theta, d). \tag{13}$$

where $\mathcal{R}(\theta, d)$ denotes the higher-order remainder term. Let the Taylor truncation error be $\mathcal{E}_{\text{Tay}}(\theta, d) \triangleq |\mathcal{R}(\theta, d)|$.

**Theorem 3.1** (Taylor truncation error). *Under Assumption 3, for any* $\theta, d$, *the Taylor truncation error satisfies* $\mathcal{E}_{Tay}(\theta, d) \leq \frac{\rho_H}{6} \|d\|_2^3$.

The Taylor truncation error is primarily governed by the update magnitude $|d|_2$, which increases under severe local drift in federated learning.

**Fisher approximation error.** FedFit uses the Fisher curvature as a surrogate for the Hessian when computing second-order scores. In practice, it employs the K-FAC approximation $\widehat{F}$. This introduces an additional bias in the second-order term: $\mathcal{E}_{\text{Fish}}(\theta, d) \triangleq \left| \frac{1}{2} d^\top \left( F(\theta) - \widehat{F}(\theta) \right) d \right|$.

**Theorem 3.2** (Fisher approximation error). *Under Assu. 4,* $\forall \theta, d$: $\mathcal{E}_{Fish}(\theta, d) \leq \frac{1}{2} \|F(\theta) - \widehat{F}(\theta)\| \|d\|_2^2 \leq \frac{1}{2} \delta_F(\theta) \|d\|_2^2$.

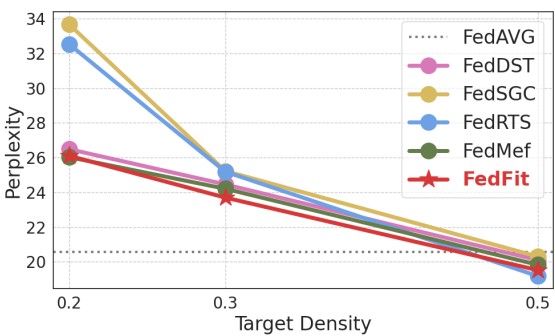

*Figure 3.* Perplexity on the TinyStories dataset using GPT-2-32M across varying target densities.

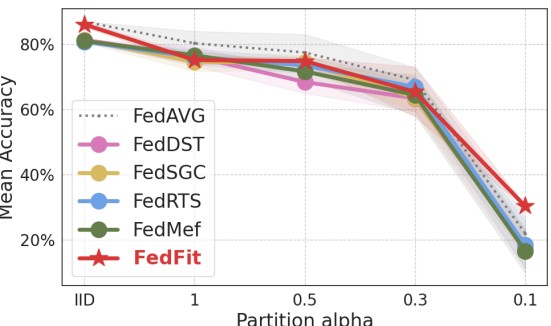

*Figure 4.* Accuracy on CIFAR-10 under varying degrees of data heterogeneity across clients.

Unlike $\mathcal{E}_{\text{Tay}}$, $\mathcal{E}_{\text{Fish}}$ is due to curvature approximation error. Even when $\|d\|_2$ is small, an overly coarse approximation $\widehat{F}$ can lead to a large $\delta_F(\theta)$ and thus induce systematic bias in the second-order scores.

**Takeaways.** Theorems 3.2 and 3.1 show that the second-order scoring errors of `FedFit` satisfy

$$\mathcal{E}_{\text{Tay}}(\theta, d) = O(\|d\|_2^3),$$
$$\mathcal{E}_{\text{Fish}}(\theta, d) = O(\delta_F(\theta)\|d\|_2^2). \tag{14}$$

This result highlights the necessity of jointly controlling the update magnitude via $\|d\|_2$ and improving the accuracy of the Fisher approximation in practice.

# 4. Evaluation

This section evaluates the performance of `FedFit` in federated learning through extensive experiments on CV and NLP benchmarks under varying data distributions and model configurations. All experiments are repeated five times with different random seeds, and we report the mean results with standard errors. Performance is summarized in figures and tables, where the best and second-best results are highlighted in **green** and yellow, respectively.

## 4.1. Experiment Setup

We benchmark `FedFit` against representative state-of-the-art (SOTA) approaches in federated learning settings for visual recognition tasks. Following common practice in FDST, in computer vision studies, we evaluate four widely used image classification datasets: (i) CIFAR-10 (Krizhevsky et al., 2009), (ii) CINIC-10 (Darlow et al., 2018), (iii) SVHN (Netzer et al., 2011) and (iv) TinyImageNet (Deng et al., 2009). We use two lightweight backbones, ResNet-18 (He et al., 2016) and ShuffleNetV2 (Zhang et al., 2018), as the default model architectures. For NLP tasks, we adopt GPT-2-32M as the backbone and evaluate on the TinyStories dataset (Eldan & Li, 2023), a benchmark designed for small language models, as detailed in Sec. D.1.

**Baselines.** We compare `FedFit` with seven representative federated learning baselines: (i) FedAvg (McMahan et al., 2017), (ii) FL-PQSU (Xu et al., 2021), (iii)FedDST (Bibikar et al., 2022), (iv) FedTiny (Huang et al., 2023), (v) Fed-Mef (Huang et al., 2024), (vi) FedSGC (Tian et al., 2024) and (vii) FedRTS (Huang et al., 2025). Collectively, these baselines cover various paradigms, including parameter averaging, federated dynamic pruning, model compression, and model fusion, allowing for a comprehensive and fair evaluation of `FedFit` in the federated learning setting, as detailed in Sec.D.1.

**Federated Learning Settings.** To simulate data heterogeneity across clients, we partition the original dataset into $N = 100$ clients through a Dirichlet sampling process with a concentration parameter $a$, where a smaller $a$ indicates stronger heterogeneity. For vision tasks, we adopt the label-distribution skew $a = 0.5$, while for unlabeled NLP tasks, we use the quantity skew $a = 5$. Federated training is conducted for $T_{\text{max}} = 500$ communication rounds, where we randomly sample 10 clients from the entire population to participate in local training and aggregation. All methods share the same local optimization procedure: in each round, every selected client performs 5 local training epochs; the batch size is set to 64 for CV tasks and 16 for NLP tasks; and we use SGD with a learning rate $\eta = 0.001$. In addition, we run an outer loop for topology adjustment once every $\Delta T = 10$ inner loop step, until the total iteration count $t$ reaches $T_{\text{end}} = 300$.

## 4.2. Performance Evaluation

**Computer Vision Tasks.** We first evaluate the performance of `FedFit` in a variety of computer vision tasks. We consider two lightweight architectures, ResNet18 and ShuffleNetV2, and conduct experiments on four datasets: CIFAR-10, CINIC-10, TinyImageNet, and SVHN. We run tests with three target densities: 30%, 20%, and 10% and report model accuracies. Table 1 summarizes the results of different methods on all datasets. `FedFit` consistently outperforms the baselines in most cases and improves roughly

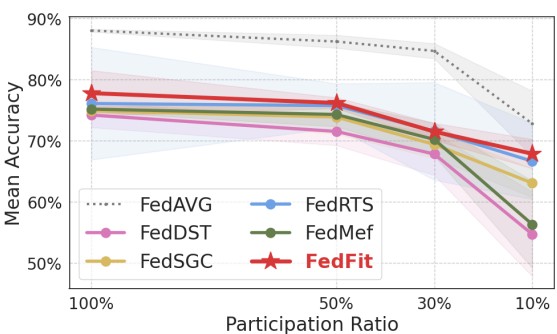

*Figure 5.* Accuracy on CIFAR-10 across different participation ratios of clients.

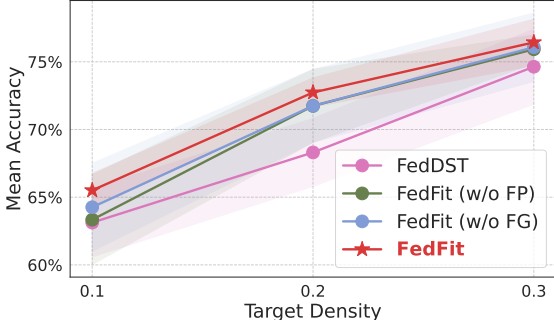

*Figure 6.* Accuracy on CIFAR-10 under different pruning and growing strategies.

3.8% over the strongest baseline on average, which suggest that the prune and grow scoring mechanism in `FedFit` can more accurately identify critical parameters, thus handling data heterogeneity more effectively in federated learning and providing better performance while maintaining high communication efficiency.

**Natural Language Processing Tasks.** To evaluate the effectiveness of FedFit across different domains, we conduct experiments on the TinyStories dataset using a GPT-2-32M model, and consider three target density levels: 50%, 30%, and 20%. We use perplexity (PPL), a standard evaluation metric for language modeling, as the primary metric, which measures how well the model assigns probability to the ground-truth text under its predictive distribution. Figure 3 demonstrates `FedFit` strong advantages and good adaptability in the NLP domain as well.

### 4.3. Robustness Analysis

To examine the stability of `FedFit` in more challenging federated environments, we experiment along two common sources of uncertainty: (i) the degree of heterogeneity in client data distributions, and (ii) the fraction of clients available for participation in each round.

**Data heterogeneity.** Non-IID data substantially increases the optimization difficulty in federated learning. We construct heterogeneous partitions of varying severity on CIFAR-10 and TinyStories, and control the distribution skew via the Dirichlet concentration parameter $a$. As shown in figure 4, even as heterogeneity becomes more pronounced, FedFit remains ahead of the baselines in most settings, demonstrating stronger adaptability to non-IID data. This advantage stems from the more reliable pruning and growth importance assessment in `FedFit`, enabling stable sparse structures under severe client heterogeneity.

**Client availability.** Insufficient client participation reduces the amount of update information available per round, which can slow convergence and increase training variance. To this end, in experiments with 10 clients, we train with par-

ticipation ratios of 100%, 50%, 30% and 10%, respectively. The figure 5 indicates that FedFit continues to outperform all baseline methods as the participation ratio decreases, and its advantage becomes more pronounced under low participation, suggesting strong robustness and adaptability in partial-participation scenarios.

### 4.4. Ablation Study

To isolate the contributions of `FedFit` under different pruning and growing strategies, we conduct ablation experiments on CIFAR-10 by comparing multiple prune-grow rule combinations. Since this ablation focuses on our scoring design, we use FedDST as the reference method because it shares the same FDST pipeline as `FedFit` while using the original prune-and-grow rules. We hypothesize that the performance of `FedFit` primarily arises from the accuracy of its pruning and growing scores.

To separately examine the roles of pruning scores and growing scores, we construct the following `FedFit` variants: (i) `FedFit` (w/o FP): we do not use Fisher-based scoring in the pruning stage, while keeping all other components unchanged. (ii) `FedFit` (w/o FG): we do not use Fisher-based scoring in the growing stage, while keeping all other components unchanged. (iii) FedDST: a reference baseline used to assess the overall effectiveness of `FedFit`, without the Fisher-based scoring design introduced in `FedFit`.

The results in Fig. 6 show that Fisher-based scoring yields more stable and consistent performance improvements. Removing Fisher pruning or Fisher growing alone leads to performance degradation to different extents, indicating that the scoring mechanisms in both stages contribute to final performance. Also, `FedFit` achieves the best results, supporting our claim that scoring accuracy is key. A comparison with FedDST further suggests that the gains in `FedFit` do not come merely from the sparse training framework itself, but from more accurate prune and grow decisions. Additional sensitivity analyses on key hyperparameters are provided in Appendix E.

## 5. Conclusion

We study FDST in cross-device federated learning and identify a key limitation of prior work: most prune and grow rules are heuristic and can be unstable under non-IID heterogeneity. To move beyond heuristics, we derive an objective-consistent importance score from a second-order loss approximation, yielding a Fisher-informed criterion for topology adjustment. To make it practical, we estimate curvature with a K-FAC Fisher approximation, avoiding explicit Hessian computation. Building on this, we propose `FedFit`, which uses theoretically grounded scores to guide structural evolution. Experiments on CV and NLP tasks show that `FedFit` learns more stable sparse structures and achieves strong accuracy and generalization with high communication efficiency and minimal extra client overhead.

## Impact Statement

This paper presents work whose goal is to advance the field of Machine Learning. There are many potential societal consequences of our work, none which we feel must be specifically highlighted here.

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

# A. Related Works

## A.1. Federated Dynamic Sparse Training

Cross-device federated learning (FL) is often constrained by on-device memory and uplink bandwidth, which motivates training sparse models directly rather than relying on post-training compression. However, the dense baseline FedAvg (McMahan et al., 2017) aggregates client updates while retaining the full cost of dense models. Towards end-to-end, PruneFL (Jiang et al., 2022) progressively refines sparse structures during federated optimization, while related works study efficient sparse training under system and data heterogeneity, including local-sparsity frameworks such as ZeroFL (Qiu et al., 2022) and inclusive pruning schemes like FedPrune (Munir et al., 2021), building on broader dynamic sparse training (DST) that co-adapts weights and sparse connectivity during training (Jayakumar et al., 2020; Raihan & Aamodt, 2020; Huang et al., 2026).

Client-driven approaches such as FedDST (Bibikar et al., 2022) and FedSGC (Tian et al., 2024) keep communication sparse, but must infer structure updates from partial, non-IID client information, which can destabilize structure evolution; server-driven variants instead leverage more global information by updating masks with gradient-derived importance (Huang et al., 2023) or fusing heterogeneous client structures for better efficiency and performance (Huang et al., 2024), at the cost of extra coordination and heavier transmitted statistics. More recently, FedRTS (Huang et al., 2025) further improves robustness under heterogeneity by probabilistic exploration with Thompson sampling. Despite these advances, most FDST frameworks still rely on heuristic criteria for pruning and reactivation which can be unreliable. For this reason, we built FedFit, enabling more stable and accurate sparse structure adaptation in FL.

## A.2. Fisher-Based Sparsity Criteria

Second-order information has a long history in pruning, with the central idea of approximating the loss locally by a quadratic expansion around a reference point and defining parameter importance by the induced loss increase after removing a parameter. Optimal Brain Damage (OBD) (LeCun et al., 1989) measures saliency using a diagonal Hessian approximation, whereas the optimal brain surgeon (OBS) (Hassibi et al., 1993) further accounts for parameter coupling through an approximate inverse Hessian to obtain a more refined second-order criterion. Structured pruning in a Kronecker-factored eigenbasis (Wang et al., 2019), efficient inverse-curvature estimation via Woodbury-style updates (Singh & Alistarh, 2020), and scalable OBS-style post-training pruning frameworks (Frantar et al., 2022). More recently, similar second-order ideas have also been adapted to one-shot pruning of massive language models (Frantar & Alistarh, 2023), while Fisher-based criteria provide another practical route to curvature-informed importance estimation (Theis et al., 2018). Importantly, these methods typically assume a post-training setting, where pruning is performed near a stationary point with relatively stable curvature. Consequently, extending them to DST or FDST with continuous rewiring introduces two key challenges: the high cost of second-order estimation and the instability of local curvature approximations during non-convergent training phases.

Recent work has revisited scalable surrogates of sensitivity and curvature. Taylor/sensitivity-based pruning has been widely adopted in practice (Molchanov et al., 2019); when explicit Hessian computation is prohibitive, Fisher information is commonly used as a tractable proxy for curvature, motivating more efficient second-order approximation methods such as WoodFisher (Singh & Alistarh, 2020) and Group Fisher Pruning (Liu et al., 2021a). In parallel, several DST variants aim to reduce the reliance on purely heuristic rewiring, for instance, by introducing trainable masked layers that make the sparse structure itself more directly optimizable (Liu et al., 2020), or by controlling structural evolution via scheduled grow-and-prune procedures (Ma et al., 2021). In federated settings, FedFisher (Jhunjhunwala et al., 2024) demonstrates the value of Fisher information for one-shot federated learning, while existing FDST methods rarely incorporate Fisher-approximated signals in a systematic manner for continual prune and grow adaptation. For this reason, we propose `FedFit`, which enables more stable and accurate sparse structure adaptation in federated learning.

## A.3. Curvature-aware Federated Optimization

Beyond pruning, curvature/second-order signals have also been explored to stabilize and accelerate federated optimization under heterogeneity. For example, FedCurv (Shoham et al., 2019) introduces a Fisher-informed quadratic regularizer to reduce client drift across non-IID data. In parallel, adaptive federated optimization (FedOpt) (Reddi et al., 2020) leverages server-side preconditioning (e.g., FedAdam/FedYogi/FedAdagrad) to improve robustness and tuning behavior under heterogeneous updates. More explicitly second-order FL methods communicate curvature surrogates to approximate Newton or quasi-Newton directions, including FedNL (Safaryan et al., 2021) and its memory and communication-efficient

extensions such as FLECS (Agafonov et al., 2025). Related Newton-type frameworks further reduce communication by approximating Newton steps in a distributed manner (Dinh et al., 2020) or by sporadically sharing low-rank curvature information. Recent quasi-Newton approaches pursue faster and sometimes fairer convergence via curvature updates at the server, such as DQN-Fed (Hamidi & Ye, 2025), and combine L-BFGS-style updates with error-feedback mechanisms under compression (Wu et al., 2024). Resource-constrained FEEL settings have also adopted Fisher-based approximations to enable distributed L-BFGS updates with low overhead (Liu et al., 2021b). While these methods focus on accelerating weight optimization, they do not address continual prune/grow decisions in FDST; in contrast, FedFit uses Fisher-approximated signals specifically to guide stable sparse structure adaptation during training.

## B. Theoretical Proofs

### B.1. Proof of Assumption 2

Directly computing the Hessian matrix is still impractical. Under Assumption 1, for the cross-entropy loss

$$\mathcal{L}(\theta) = \mathbb{E}_{z \sim D}[-\log Q_\theta(z \mid I)], \tag{15}$$

the Hessian matrix H is the second derivative of the loss with respect to $\theta$. Specifically,

$$H = \nabla_\theta^2 \mathcal{L}(\theta) = \nabla_\theta^2 \mathbb{E}_{z \sim D}[-\log Q_\theta(z \mid I)]. \tag{16}$$

We can move the second-order derivative inside the integral and summation operator to obtain

$$H = \mathbb{E}_{z \sim D}\left[-\nabla_\theta^2 \log Q_\theta(z \mid I)\right]. \tag{17}$$

Under Assumption A2, we replace the data distribution $D$ with the model distribution $Q_\theta$. In this case, the expected negative Hessian of the log-likelihood coincides with the expected Fisher information:

$$F \triangleq -\mathbb{E}_{z \sim Q_\theta}\left[\nabla_\theta^2 \log Q_\theta(z \mid I)\right], \tag{18}$$

and thus $H = F$ holds in expectation under $Q_\theta$. In general, $H$ and $F$ may differ, throughout the paper we use Fisher as a practical curvature surrogate $H \approx F$. Here $F$ denotes the expected Fisher under $Q_\theta$ used only to motivate the Hessian-Fisher connection. In implementation, we rely on the empirical Fisher computed from client data and its K-FAC-based approximation as a practical surrogate for curvature.

### B.2. Derivation of K-FAC Approximation

Let the model parameters be $\theta = \{W_\ell\}_{\ell=1}^L$, and the loss function be $\mathcal{L}(\theta; x, y)$. The empirical Fisher is defined as the expectation of the outer product of per-sample gradients:

$$F_{\text{emp}}(\theta) = \mathbb{E}_{(x,y) \sim \mathcal{D}}\left[\nabla_\theta \mathcal{L}(\theta; x, y) \nabla_\theta \mathcal{L}(\theta; x, y)^\top\right]. \tag{19}$$

After partitioning parameters by layers, $F_{\text{emp}}$ can be written as a block matrix, whose $(\ell, m)$-th block is

$$F_{\ell m} = \mathbb{E}\left[\text{vec}(\nabla_{W_\ell} \mathcal{L}) \text{vec}(\nabla_{W_m} \mathcal{L})^\top\right]. \tag{20}$$

For layer $\ell$, we have

$$s_\ell = W_\ell a_\ell, \tag{21}$$

where $a_\ell \in \mathbb{R}^{d_{\text{in},\ell}}$ denotes the input activation to this layer, $s_\ell$ denotes the pre-activation, and $W_\ell \in \mathbb{R}^{d_{\text{out},\ell} \times d_{\text{in},\ell}}$ is the parameter matrix. Let

$$\delta_\ell = \nabla_{s_\ell} \mathcal{L} \in \mathbb{R}^{d_{\text{out},\ell}} \tag{22}$$

be the backpropagated gradient with respect to the pre-activation. By the chain rule, the parameter gradient for this layer satisfies

$$\nabla_{W_\ell} \mathcal{L} = \delta_\ell a_\ell^\top. \tag{23}$$

Vectorizing (23) and using the identity $\text{vec}(uv^\top) = v \otimes u$, we obtain

$$\text{vec}(\nabla_{W_\ell}\mathcal{L}) \; = \; \text{vec}(\delta_\ell a_\ell^\top) \; = \; a_\ell \otimes \delta_\ell. \tag{24}$$

Substituting (24) into the diagonal block of (20) (i.e., $m = \ell$) yields

$$F_{\ell\ell} \; = \; \mathbb{E}\Big[(a_\ell \otimes \delta_\ell)(a_\ell \otimes \delta_\ell)^\top\Big] \; = \; \mathbb{E}\Big[(a_\ell a_\ell^\top) \otimes (\delta_\ell \delta_\ell^\top)\Big]. \tag{25}$$

To obtain an efficient curvature approximation, K-FAC adopts the following two standard assumptions.

**(i) Cross-layer block-diagonal approximation.** K-FAC ignores correlations between parameter gradients across different layers and sets the off-diagonal Fisher blocks to zero:

$$F_{\text{emp}}(\theta) \; \approx \; \text{blockdiag}(F_{11}, F_{22}, \dots, F_{LL}). \tag{26}$$

**(ii) Intra-layer Kronecker-factored approximation.** For each layer block (25), K-FAC assumes that the activations and backpropagated gradients are independent in terms of second-order statistics, so that

$$F_{\ell\ell} = \mathbb{E}\Big[(a_\ell a_\ell^\top) \otimes (\delta_\ell \delta_\ell^\top)\Big] \; \approx \; \mathbb{E}[a_\ell a_\ell^\top] \otimes \mathbb{E}[\delta_\ell \delta_\ell^\top] \; = \; A_\ell \otimes G_\ell, \tag{27}$$

where

$$A_\ell = \mathbb{E}[a_\ell a_\ell^\top] \in \mathbb{R}^{d_{\text{in},\ell} \times d_{\text{in},\ell}}, \qquad G_\ell = \mathbb{E}[\delta_\ell \delta_\ell^\top] \in \mathbb{R}^{d_{\text{out},\ell} \times d_{\text{out},\ell}}. \tag{28}$$

Combining (26) and (27) gives the K-FAC approximation used in this paper. *(Note: $G_\ell$ here is the K-FAC factor for layer $\ell$ and should not be confused with the growth score notation in the main text.)*

The above derivation holds for fully connected layers. For convolutional layers, one can view a convolution as a linear mapping over local input patches: by unfolding patches, the convolution can be written as $s_\ell = W_\ell \phi(a_\ell)$. Letting $a_\ell$ denote the unfolded patch vector then leads to the same derivation.

### B.3. Derivation of growing score

Accordingly, the predicted loss decrease induced by activating the $j$-th parameter can be written as

$$q_j(\alpha) = \mathcal{L}(\theta) - \mathcal{L}(\theta + \alpha e_j), \tag{29}$$

and is approximated by

$$q_j(\alpha) \approx -g_j\alpha - \frac{1}{2}H_{jj}\alpha^2. \tag{30}$$

In federated learning, explicitly computing Hessian matrix is expensive, so we approximate the Hessian entry using the empirical Fisher information:

$$H \approx F_{\text{emp}}. \tag{31}$$

The empirical Fisher is defined as

$$F_{\text{emp}}(\theta) = \mathbb{E}_{z \sim D}\big[s_\theta(z)s_\theta(z)^\top\big], \tag{32}$$

where the score function is

$$s_\theta(z) = \nabla_\theta \log Q_\theta(z). \tag{33}$$

Since for any vector $v$ we have

$$v^\top F_{\text{emp}} v = \mathbb{E}_{z \sim D}\big[(v^\top s_\theta(z))^2\big] \geq 0, \tag{34}$$

$F_{\text{emp}}$ is positive semidefinite, and thus its diagonal entries are nonnegative, which facilitates numerically stable implementation. In practice, we add a small damping term in the denominator to avoid division by zero. Therefore, the above expression is a concave quadratic function of $\alpha$, and the optimal step size can be obtained by differentiation:

$$\frac{dq_j(\alpha)}{d\alpha} = -g_j - F_{jj}\alpha, \tag{35}$$

which gives the optimal growth step size

$$\alpha_j^\star = -\frac{g_j}{F_{jj}}. \tag{36}$$

Substituting Eq. 36 into Eq. 30 after applying Eq. 31 yields the maximum predicted decrease

$$\Delta_j = q_j(\alpha_j^\star) = \frac{1}{2}\frac{g_j^2}{F_{jj}}. \tag{37}$$

Finally, the single-parameter growth score is

$$\mathcal{G}_j = \frac{1}{2}\frac{g_j^2}{(F_{\text{emp}})_{jj}}. \tag{38}$$

We therefore prioritize growing the parameters with larger $\mathcal{G}_n$.

### B.4. Layerwise Pruning Analysis

We analyze the second-order loss change induced by pruning multiple parameters within a fixed layer $\ell$. Let the parameter vector of this layer be $\theta_\ell$, and let $S_\ell$ denote the index set of parameters to be pruned. We define the corresponding pruning perturbation $d_\ell$ coordinate-wise as

$$(d_\ell)_i = \begin{cases} -\theta_i, & i \in S_\ell, \\ 0, & i \notin S_\ell, \end{cases} \tag{39}$$

which zeros out all coordinates indexed by $S_\ell$.

Under Assumption 1, a second-order Taylor expansion of the objective with respect to the layer parameters yields

$$\mathcal{L}(\theta_\ell + d_\ell) - \mathcal{L}(\theta_\ell) \approx g_\ell^\top d_\ell + \frac{1}{2}d_\ell^\top H_\ell d_\ell, \tag{40}$$

where

$$\begin{aligned} g_\ell &= \nabla_{\theta_\ell}\mathcal{L}(\theta_\ell), \\ H_\ell &= \nabla_{\theta_\ell}^2\mathcal{L}(\theta_\ell). \end{aligned} \tag{41}$$

Consistent with Eq. (2) in main text, we further approximate $H_\ell \approx F_\ell$ and obtain

$$\Delta\mathcal{L}_\ell(S_\ell) := \mathcal{L}(\theta_\ell + d_\ell) - \mathcal{L}(\theta_\ell) \approx g_\ell^\top d_\ell + \frac{1}{2}d_\ell^\top F_\ell d_\ell. \tag{42}$$

We next expand the two terms in Eq. (42). By Eq. (39), the linear term reduces to

$$g_\ell^\top d_\ell = \sum_i g_i(d_\ell)_i = \sum_{i \in S_\ell} g_i(-\theta_i) = -\sum_{i \in S_\ell} g_i\theta_i. \tag{43}$$

For the quadratic term, we have

$$d_\ell^\top F_\ell d_\ell = \sum_i \sum_j (d_\ell)_i(F_\ell)_{ij}(d_\ell)_j. \tag{44}$$

Since $(d_\ell)_i = 0$ whenever $i \notin S_\ell$, the summation can be restricted to indices in $S_\ell$:

$$\begin{aligned} d_\ell^\top F_\ell d_\ell &= \sum_{i \in S_\ell}\sum_{j \in S_\ell}(d_\ell)_i(F_\ell)_{ij}(d_\ell)_j \\ &= \sum_{i \in S_\ell}\sum_{j \in S_\ell}(-\theta_i)(F_\ell)_{ij}(-\theta_j) \\ &= \sum_{i \in S_\ell}\sum_{j \in S_\ell}\theta_i(F_\ell)_{ij}\theta_j. \end{aligned} \tag{45}$$

Separating diagonal and off-diagonal contributions gives

$$d_\ell^\top F_\ell d_\ell = \sum_{i \in S_\ell} \theta_i^2 (F_\ell)_{ii} + \sum_{i \neq j \in S_\ell} \theta_i (F_\ell)_{ij} \theta_j. \tag{46}$$

Substituting Eq. (43) and Eq. (46) into Eq. (42) yields the second-order approximation of the layerwise loss increment:

$$\Delta \mathcal{L}_\ell(S_\ell) \approx \sum_{i \in S_\ell} \left( -g_i \theta_i + \frac{1}{2} \theta_i^2 (F_\ell)_{ii} \right) + \frac{1}{2} \sum_{i \neq j \in S_\ell} \theta_i (F_\ell)_{ij} \theta_j. \tag{47}$$

The first term aggregates the single-parameter pruning signals, while the second term captures intra-layer interactions induced by off-diagonal Fisher entries. We denote this interaction residual by

$$\mathcal{R}_\ell(S_\ell) = \frac{1}{2} \sum_{i \neq j \in S_\ell} \theta_i (F_\ell)_{ij} \theta_j. \tag{48}$$

Directly optimizing Eq. (47) over $S_\ell$ is a quadratic combinatorial problem due to $\mathcal{R}_\ell(S_\ell)$, which is costly for large-scale networks and especially in federated settings. Therefore, we adopt a separable surrogate

$$\widetilde{\Delta \mathcal{L}}_\ell(S_\ell) = \sum_{i \in S_\ell} \Delta_i, \tag{49}$$

and select $S_\ell$ by approximating $\Delta \mathcal{L}_\ell(S_\ell)$ with $\widetilde{\Delta \mathcal{L}}_\ell(S_\ell)$. Under a fixed target density constraint $\rho$, let the number of parameters in layer $\ell$ be $d_\ell$, and let the pruning budget be $m_\ell = (1 - \rho)d_\ell$. Since the surrogate objective is separable, minimizing $\widetilde{\Delta \mathcal{L}}_\ell(S_\ell)$ is equivalent to pruning the $m_\ell$ parameters with the smallest $\Delta_i$ within the layer.

Finally, we characterize the magnitude of the interaction residual $\mathcal{R}_\ell(S_\ell)$. Define the diagonal and off-diagonal components of the layer Fisher as

$$D_\ell = \text{diag}(F_\ell), \qquad R_\ell = F_\ell - D_\ell. \tag{50}$$

Let $\theta_{S_\ell}$ be the subvector of $\theta_\ell$ restricted to indices in $S_\ell$. Then $\mathcal{R}_\ell(S_\ell)$ can be written in matrix form as

$$\mathcal{R}_\ell(S_\ell) = \frac{1}{2} \theta_{S_\ell}^\top (R_\ell)_{S_\ell} \theta_{S_\ell}. \tag{51}$$

By standard matrix norm inequalities, we obtain

$$\left| \mathcal{R}_\ell(S_\ell) \right| \leq \frac{1}{2} \left\| (R_\ell)_{S_\ell} \right\|_2 \left\| \theta_{S_\ell} \right\|_2^2 \leq \frac{1}{2} \left\| R_\ell \right\|_2 \left\| \theta_{S_\ell} \right\|_2^2. \tag{52}$$

The residual increases with stronger off-diagonal Fisher interactions (larger $\|R_\ell\|_2$) and larger pruning perturbations (larger $\|\theta_{S_\ell}\|_2$). Thus, when $\|R_\ell\|_2$ is moderate and the selected set yields a small $\|\theta_{S_\ell}\|_2$, the interaction residual is controlled. Moreover, our layerwise selection rule sorting by $\Delta_i$ and pruning the smallest ones tends to remove parameters with small $\theta_i^2 (F_\ell)_{ii}$ and $|g_i \theta_i|$, which in turn favors a smaller $\|\theta_{S_\ell}\|_2$ and tightens the bound in Eq. (52).

### B.5. Layerwise Growing Analysis

We provide a layerwise counterpart of the single-parameter growing rule in Sec. 3.3. Consider a fixed layer $\ell$ with parameter vector $\theta_\ell$. Let $G_\ell$ denote the index set of parameters to be activated in this layer. We apply a growing perturbation $d_\ell$ to the layer parameters, defined coordinate-wise as

$$(d_\ell)_i = \begin{cases} \alpha_i, & i \in G_\ell, \\ 0, & i \notin G_\ell, \end{cases} \tag{53}$$

where $\alpha_i$ is the growth step size associated with coordinate $i$.

Under Assumption 1, applying a second-order Taylor expansion to the layer objective gives

$$\mathcal{L}(\theta_\ell + d_\ell) - \mathcal{L}(\theta_\ell) \approx g_\ell^\top d_\ell + \frac{1}{2} d_\ell^\top H_\ell d_\ell, \tag{54}$$

where $g_\ell = \nabla_{\theta_\ell} \mathcal{L}(\theta_\ell)$ and $H_\ell = \nabla^2_{\theta_\ell} \mathcal{L}(\theta_\ell)$. Consistent with Eq. (2), we further approximate $H_\ell \approx F_\ell$ and obtain

$$\Delta\mathcal{L}_\ell(G_\ell) := \mathcal{L}(\theta_\ell + d_\ell) - \mathcal{L}(\theta_\ell) \approx g_\ell^\top d_\ell + \frac{1}{2} d_\ell^\top F_\ell d_\ell. \tag{55}$$

Accordingly, the predicted *loss decrease* induced by growing $G_\ell$ is

$$\mathcal{Q}_\ell(G_\ell) := \mathcal{L}(\theta_\ell) - \mathcal{L}(\theta_\ell + d_\ell) \approx -\Delta\mathcal{L}_\ell(G_\ell). \tag{56}$$

We now expand the two terms in Eq. (55). By Eq. (53), the linear term becomes

$$g_\ell^\top d_\ell = \sum_i g_i (d_\ell)_i = \sum_{i \in G_\ell} g_i \alpha_i. \tag{57}$$

The quadratic form can be written as

$$d_\ell^\top F_\ell d_\ell = \sum_i \sum_j (d_\ell)_i (F_\ell)_{ij} (d_\ell)_j. \tag{58}$$

Since $(d_\ell)_i = 0$ whenever $i \notin G_\ell$, the summation can be restricted to $G_\ell$:

$$\begin{aligned} d_\ell^\top F_\ell d_\ell &= \sum_{i \in G_\ell} \sum_{j \in G_\ell} (d_\ell)_i (F_\ell)_{ij} (d_\ell)_j \\ &= \sum_{i \in G_\ell} \sum_{j \in G_\ell} \alpha_i (F_\ell)_{ij} \alpha_j. \end{aligned} \tag{59}$$

Separating diagonal and off-diagonal terms gives

$$d_\ell^\top F_\ell d_\ell = \sum_{i \in G_\ell} \alpha_i^2 (F_\ell)_{ii} + \sum_{i \neq j \in G_\ell} \alpha_i (F_\ell)_{ij} \alpha_j. \tag{60}$$

Substituting Eq. (57) and Eq. (60) back into Eq. (55), we obtain

$$\Delta\mathcal{L}_\ell(G_\ell) \approx \sum_{i \in G_\ell} \left( g_i \alpha_i + \frac{1}{2} \alpha_i^2 (F_\ell)_{ii} \right) + \frac{1}{2} \sum_{i \neq j \in G_\ell} \alpha_i (F_\ell)_{ij} \alpha_j. \tag{61}$$

Equivalently, the predicted loss decrease is

$$\mathcal{Q}_\ell(G_\ell) \approx \sum_{i \in G_\ell} \left( - g_i \alpha_i - \frac{1}{2} \alpha_i^2 (F_\ell)_{ii} \right) - \frac{1}{2} \sum_{i \neq j \in G_\ell} \alpha_i (F_\ell)_{ij} \alpha_j. \tag{62}$$

The first part aggregates the single-parameter growing gains, while the second part captures intra-layer interactions induced by off-diagonal Fisher entries. We denote the interaction residual by

$$\mathcal{R}_\ell^{\text{grow}}(G_\ell) = \frac{1}{2} \sum_{i \neq j \in G_\ell} \alpha_i (F_\ell)_{ij} \alpha_j. \tag{63}$$

Optimizing Eq. (62) jointly over $(G_\ell, \{\alpha_i\}_{i \in G_\ell})$ involves a quadratic combinatorial problem due to $\mathcal{R}_\ell^{\text{grow}}(G_\ell)$, which is expensive in large-scale networks and federated settings. Therefore, we adopt a separable surrogate that ignores off-diagonal interactions and optimizes each coordinate independently. Concretely, for a fixed index $i$, the diagonal surrogate of Eq. (62) yields the concave quadratic function

$$\widetilde{\mathcal{Q}}_i(\alpha_i) = -g_i \alpha_i - \frac{1}{2} \alpha_i^2 (F_\ell)_{ii}. \tag{64}$$

Maximizing $\widetilde{\mathcal{Q}}_i(\alpha_i)$ gives the optimal step size

$$\alpha_i^\star = -\frac{g_i}{(F_\ell)_{ii}}, \tag{65}$$

and the corresponding single-parameter growth gain

$$\mathcal{G}_i = \frac{1}{2}\frac{g_i^2}{(F_\ell)_{ii}}. \tag{66}$$

Under a fixed target density constraint $\rho$, let the number of parameters in layer $\ell$ be $d_\ell$, so that the grow budget is $m_\ell = (1 - \rho)d_\ell$. Using the separable gains in Eq. (66), we form $G_\ell$ by selecting the $m_\ell$ parameters with the largest $\mathcal{G}_i$ in that layer.

Finally, we bound the interaction residual $\mathcal{R}_\ell^{\text{grow}}(G_\ell)$. Using the diagonal and off-diagonal decomposition of the layer Fisher,

$$D_\ell = \text{diag}(F_\ell), \qquad R_\ell = F_\ell - D_\ell, \tag{67}$$

and letting $\alpha_{G_\ell}$ denote the subvector of $\{\alpha_i\}$ restricted to indices in $G_\ell$, we can write

$$\mathcal{R}_\ell^{\text{grow}}(G_\ell) = \frac{1}{2}\alpha_{G_\ell}^\top (R_\ell)_{G_\ell}\,\alpha_{G_\ell}. \tag{68}$$

By standard matrix norm inequalities, we have

$$\left|\mathcal{R}_\ell^{\text{grow}}(G_\ell)\right| \le \frac{1}{2}\left\|(R_\ell)_{G_\ell}\right\|_2 \left\|\alpha_{G_\ell}\right\|_2^2 \le \frac{1}{2}\left\|R_\ell\right\|_2 \left\|\alpha_{G_\ell}\right\|_2^2. \tag{69}$$

Therefore, the interaction residual increases with stronger off-diagonal Fisher interactions (larger $\|R_\ell\|_2$) and larger growth perturbations (larger $\|\alpha_{G_\ell}\|_2$), as suggested by Eq. (69). Consequently, when $\|R_\ell\|_2$ is moderate and $\|\alpha_{G_\ell}\|_2$ remains bounded, the interaction residual is controlled.

### B.6. Computation overhead

For a layer $\ell$, K-FAC approximates the Fisher block as $F_\ell \approx A_\ell \otimes G_\ell$. Instead of forming the full matrices $A_\ell$ and $G_\ell$, we only compute their diagonals, i.e., $\text{diag}(A_\ell) = \mathbb{E}[a_\ell \odot a_\ell]$ and $\text{diag}(G_\ell) = \mathbb{E}[g_\ell \odot g_\ell]$, where $a_\ell$ and $g_\ell$ denote the layer activations and backpropagated gradients, respectively. Given a mini-batch of size $B$, computing these diagonals costs $\mathcal{O}(B\,d_{\text{in},\ell})$ and $\mathcal{O}(B\,d_{\text{out},\ell})$, where $d_{\text{in},\ell}$ and $d_{\text{out},\ell}$ are the input and output dimensions of layer $\ell$. Thus, the additional Fisher-related cost is $\mathcal{O}(B\sum_\ell(d_{\text{in},\ell} + d_{\text{out},\ell}))$, which is lightweight compared to standard training.

For growth scoring, computing $\mathcal{G}_j$ requires the gradient $g_j$ and the diagonal Fisher entry $(F_\ell)_{jj}$ at inactive coordinates. These quantities are obtained from the same forward and backward passes used for local training, and we only materialize the required diagonal statistics as scalars, rather than constructing or storing any dense Fisher matrix.

### B.7. Memory Overhead

In terms of memory complexity, FedFit introduces little additional memory compared with prior sparse training methods. The activations required for Fisher-based score computation are already retained during standard backpropagation and can be reused directly. Therefore, the extra memory mainly comes from implementation-level factors, such as hooks for collecting layerwise statistics, rather than from storing dense Fisher matrices.

To empirically verify this, we measure client peak memory usage on a resource-constrained device equipped with an RTX 3080 GPU. As shown in Table 2, FedFit has comparable client peak memory to FedDST, FedTiny, FedMef, and FedRTS, and remains lower than FedSGC. Since FedFit is developed based on FedDST, the direct comparison with FedDST shows that our Fisher-guided scoring mechanism introduces only 0.7 MiB additional peak memory.

### B.8. Communication Cost

Let $O_d$ and $O_s$ denote the storage cost of dense and sparse parameters, respectively. In the inner loop, all federated pruning methods except FedAvg share the same communication cost, equal to $2O_s$. In the outer loop, different federated frameworks incur different communication costs, summarized as follows:

- **FedAvg:** Each round exchanges $2O_d$, accounting for both download and upload of dense parameters.

- **FedDST:** The mask does not require additional storage beyond the sparse representation, so the per-round exchange is $2O_s$.

*Table 2.* Client peak memory usage of different methods.

| Method | Client Peak Memory (MiB) |
|--------|--------------------------|
| FedDST | 24.9 |
| FedTiny | 24.9 |
| FedMef | 24.2 |
| FedSGC | 30.9 |
| FedRTS | 24.9 |
| FedFit | 25.6 |

- **PruneFL:** Clients additionally upload full-size squared gradients for adjustment, yielding an outer-loop cost of $O_d + O_s$.

- **FedSGC:** An extra gradient-congruity signal (with the same size as sparse weights) is transmitted each round, leading to $3O_s$.

- **FedTiny and FedMef:** Compared with FedDST, these methods upload TopK gradients every $\Delta R$ rounds; hence the maximum per-round exchange is $2O_s + O_\xi$, where $O_\xi$ is the storage for TopK gradients.

- **FedRTS:** FedRTS only uploads the indices of the TopK gradients every $\Delta R$ rounds. The TopK budget is denoted by $\xi_t$ and scales with the number of currently active parameters $n_\theta$. Concretely, $\xi_t = \zeta_t(1 - s_m)n_\theta$, where $\zeta_t = 0.2\left(1 + \cos\frac{t\pi}{R_{\text{stop}}E}\right)$ controls the adjustment rate at iteration $t$. Consequently, the maximum data exchange per round is $2O_s + 0.5O_\xi$, where $O_\xi$ denotes the storage associated with TopK-gradient-scale auxiliary signals.

- **FedFit:** FedFit follows the same communication pattern as FedDST. The Fisher-based prune and grow scores are used only for local topology adjustment on clients and are not transmitted to the server. Thus, clients only exchange sparse model parameters, and the per-round communication cost remains $2O_s$.

## C. Framework

### C.1. FedFit

The framework of FedFit initializes the global weights $W^0$ and the sparse mask $m^0$ on the server, and sets the target density $\rho$. It then specifies the hyperparameters related to structure adjustment, including the adjustment interval $\Delta R$ and the stopping round $R_{\text{end}}$. Here, $\Delta R$ controls how frequently structure adjustment is performed, while $R_{\text{end}}$ restricts structure adjustment to the early/mid stage of training to avoid introducing extra perturbations near convergence.

Training proceeds for a total of $R$ rounds. In each round $r$, the server randomly samples a participating set $C_r$ from all clients $N$ and broadcasts the current global model $(W^r, m^r)$ to all selected clients. After receiving $(W^r, m^r)$, each client $c \in C_r$ performs $E$ epochs of local training on its dataset $D_c$, updating weights only on the current sparse subnetwork via $W_c \leftarrow W_c - \eta\nabla f_c(W_c \odot m_c; B)$, where $B$ denotes a local mini-batch and $\odot$ is the element-wise masking operator.

When the structure adjustment condition is met, FedFit triggers a Fisher-based score computation on the client side. Each client estimates the Fisher information using its current local model and $D_c$, producing a score tensor $s_c$ with the same shape as the parameters. The client then performs a local score-based rewire according to $s_c$: under the constraint that the target density $\rho$ is preserved, it prunes low-score connections and grows new connections at high-score candidate locations, thereby enabling local structural evolution with *fixed density but changing structure*. Finally, the client uploads the updated weights and structural information for this round and also uploads the score $s_c$ for server-side aggregation.

After receiving uploads from all participating clients, the server first aggregates the weights to obtain $W^{r+\frac{1}{2}} = \sum_{c \in C_r} p_c W_c$, where $p_c$ is typically proportional to the number of samples on client $c$. If the current round is a structure adjustment round and $r < R_{\text{end}}$, the server similarly aggregates the uploaded scores to obtain a global score $s^{r+\frac{1}{2}} = \sum_{c \in C_r} p_c s_c$. Based on the aggregated score $s^{r+\frac{1}{2}}$, the server performs a further score-based pruning step on the current global mask to enforce the target density $\rho$, yielding $m^{r+1}$. Otherwise, the server keeps the mask unchanged, i.e., $m^{r+1} = m^r$. The server then sets $W^{r+1} = W^{r+\frac{1}{2}}$ and proceeds to the next round.

After $R$ rounds, the server outputs the final sparse model $(W^R, m^R)$. Overall, FedFit uses Fisher scores on the client side to perform local rewiring that adapts to structural preferences induced by non-IID data, while on the server side it aggregates

---

**Algorithm 1** Overview of FedFit

---

**Input:** Clients $N$ with local datasets $D_i$; target density $\rho$; learning rate $\eta$; update schedule $\Delta R, R_{\text{end}}, \alpha^r$.
**Output:** Final sparse model $(W^R, m^R)$.
Initialize server model $(W^0, m^0)$ with $\|m^0\|_0 = \rho$;
**for** $r = 0$ **to** $R - 1$ **do**
    Sample clients $C_r \subset [N]$;
    Transmit server model $(W^r, m^r)$ to all $c \in C_r$;
    **for all** each client $c \in C_r$ **in parallel do**
        Receive $(W_c, m_c) \leftarrow (W^r, m^r)$;
        **for** $e = 1$ **to** $E$ **do**
            Sample minibatch $B$ from $D_c$;
            Update $W_c \leftarrow W_c - \eta \nabla f_c(W_c \odot m_c; B)$;
            **if** $r \bmod \Delta R = 0$ **and** $e = E$ **and** $r < R_{\text{end}}$ **then**
                Compute Fisher-based scores: $s_c \leftarrow$ FISHERSCORE$(W_c, m_c; D_c)$;
                Score-based pruning: $m_c \leftarrow$ PRUNEBYSCORE$(m_c, s_c; \rho)$;
                Score-based growth: $m_c \leftarrow$ GROWBYSCORE$(m_c, s_c; \rho)$;
            **end if**
        **end for**
        Transmit $(W_c, m_c, s_c)$ to server;
    **end for**
    Aggregate weights $W^{r+\frac{1}{2}} \leftarrow \sum_{c \in C_r} p_c W_c$;
    **if** $r \bmod \Delta R = 0$ **and** $r < R_{\text{end}}$ **then**
        Aggregate scores $s^{r+\frac{1}{2}} \leftarrow \sum_{c \in C_r} p_c s_c$;
        Server score-based pruning: $(W^{r+1}, m^{r+1}) \leftarrow$ PRUNEBYSCORE$(m^r, s^{r+\frac{1}{2}}; \rho)$;
    **else**
        $(W^{r+1}, m^{r+1}) \leftarrow (W^{r+\frac{1}{2}}, m^r)$;
    **end if**
**end for**

---

scores and applies an additional global pruning step to unify and stabilize the global structure, thereby enabling more robust and consistent sparse training in federated settings.

## D. Additional Experiments setup

### D.1. Baselines

The selected baselines differ in their approaches to federated optimization and sparse structure learning:

- **FedAvg:** Performs standard federated optimization by averaging client model updates on the server, and serves as the primary dense reference baseline in all experiments.

- **FL-PQSU:** Integrates structured pruning, weight quantization, and selective updating into the federated training process to reduce computation and communication overhead while preserving model accuracy.

- **FedDST:** Applies dynamic sparse training on clients and periodically aggregates sparse updates to form a new global sparse structure, enabling structure evolution throughout training.

- **FedTiny:** Maintains a compact sparse model by updating the mask based on importance signals derived from client gradients, aiming to improve efficiency while preserving accuracy.

- **FedMef:** Improves global performance by assembling and fusing heterogeneous local models on the server, which allows clients to train personalized structures while still benefiting from global aggregation.

- **FedSGC:** Adjusts model topology on devices guided by the gradient congruity, which is similar to FedDST but requires extra communication cost.

*Table 3.* Sensitivity analysis of `FedFit` with respect to the growth ratio on CIFAR-10.

| Density | 0.1 | 0.2 | 0.3 |
|---------|-----|-----|-----|
| 0.1 | $65.60 \pm 2.36$ | $65.51 \pm 1.25$ | $65.50 \pm 3.36$ |
| 0.2 | $73.75 \pm 2.34$ | $74.43 \pm 1.07$ | $74.10 \pm 1.05$ |
| 0.3 | $75.79 \pm 2.15$ | $76.43 \pm 1.72$ | $75.12 \pm 2.75$ |

- **FedRTS:** Uses Thompson Sampling to stably adjust sparse structures under heterogeneity, yielding more robust sparse training and improved accuracy in non-IID environments.

## D.2. Datasets

Following common practice in federated computer vision studies, we evaluate `FedFit` on four widely used image classification benchmarks: CIFAR-10, CINIC-10, SVHN, and TinyImageNet.

- **CIFAR-10:** It contains 50,000 training images and 10,000 testing images. Each image is a $32 \times 32$ RGB image, covering 10 object classes.

- **CINIC-10:** It contains three equal subsets (train/validation/test), each comprising 90,000 images. CINIC-10 extends CIFAR-10 by including additional images collected from ImageNet.

- **SVHN:** It contains 73,257 digit images for training and 26,032 images for testing. The digit images are obtained from house numbers in Google Street View images.

- **TinyImageNet:** It contains 200 classes with 500 training images, 50 validation images, and 50 test images per class, where all images are resized to $64 \times 64$ pixels.

In the NLP tasks, we use GPT-2-32M as the backbone model and evaluate on the TinyStories dataset. TinyStories consists of synthetically generated short stories with a limited vocabulary. We truncate each story to 256 tokens, using 2,120,000 stories for training and 22,000 for testing.

## E. Hyperparameter Analysis

We conduct additional sensitivity analyses on key hyperparameters using ResNet18 on CIFAR-10. Specifically, we study the impact of the growth ratio, the topology adjustment interval $\Delta T$, and the number of local epochs.

### E.1. The Impact of Growth Ratio

The growth ratio controls the fraction of sparse connections that are pruned and regrown during each topology adjustment. A small growth ratio may slow down topology exploration, while an overly large growth ratio may introduce unstable structural changes. As shown in Table 3, `FedFit` is fairly robust to the growth ratio across different target densities. The default value of $0.2$ achieves the best or near-best performance in most cases, suggesting that a moderate growth ratio provides a good balance between topology exploration and training stability.

### E.2. The Impact of Topology Adjustment Interval $\Delta T$

The topology adjustment interval $\Delta T$ determines how frequently the sparse structure is updated. A smaller $\Delta T$ enables more frequent topology adaptation but may increase the instability caused by repeated structural changes, while a larger $\Delta T$ allows longer weight optimization between two topology updates but may delay the discovery of better sparse structures. As reported in Table 4, the performance variation under different $\Delta T$ values remains modest. This indicates that `FedFit` is reasonably robust to the adjustment interval, and we use $\Delta T = 10$ as the default setting in our experiments.

### E.3. The Impact of Local Epochs

The number of local epochs controls the amount of client-side optimization before aggregation. Increasing local epochs can improve local optimization, but excessive local training may also amplify client drift under non-IID data and make the estimated importance scores more client-specific. As shown in Table 5, using more local epochs does not consistently

*Table 4.* Sensitivity analysis of `FedFit` with respect to the topology adjustment interval $\Delta T$ on CIFAR-10.

| Density | 3 | 5 | 10 | 20 |
|---------|---|---|----|----|
| 0.1 | $65.48 \pm 2.73$ | $63.56 \pm 1.37$ | $65.51 \pm 1.25$ | $66.89 \pm 2.61$ |
| 0.2 | $74.26 \pm 1.13$ | $74.40 \pm 1.93$ | $74.43 \pm 1.07$ | $74.66 \pm 2.94$ |
| 0.3 | $76.33 \pm 1.76$ | $75.80 \pm 1.12$ | $76.43 \pm 1.72$ | $75.75 \pm 2.29$ |

*Table 5.* Sensitivity analysis of `FedFit` with respect to the number of local epochs on CIFAR-10. All results are reported as mean±std.

| Density | 5 | 10 | 20 |
|---------|---|----|----|
| 0.1 | $65.51 \pm 1.25$ | $64.11 \pm 2.70$ | $62.90 \pm 2.00$ |
| 0.2 | $74.43 \pm 1.07$ | $73.20 \pm 4.42$ | $74.65 \pm 2.10$ |
| 0.3 | $76.43 \pm 1.72$ | $76.19 \pm 2.84$ | $75.59 \pm 1.61$ |

improve performance and can even degrade accuracy at some densities. Therefore, we use $E = 5$ as the default setting, which provides a more stable trade-off between local optimization and global sparse structure learning.

