# OpenReview forum: "FedFit: Federated Dynamic Sparse Training via Fisher Information scoring"
_ICML.cc/2026/Conference — ICML 2026 regular_

### Official Review · Reviewer_jwVK · 2026-02-20

**Soundness:** 3
**Presentation:** 4
**Significance:** 2
**Originality:** 3
**Overall Recommendation:** 5
**Confidence:** 3

**Summary:**

The authors propose a sparse federated learning framework, that uses a Taylor expansion to approximate loss impact of parameters to decide if they should be enabled or disabled. Different to most Fisher based works in compression, they explicitly do not discard the linear term, which is theoretically well motivated as the model is not converged. They further Performance improves over the baseline in a wide range of tasks supporting the validity of the approach.

**Compliance With Llm Reviewing Policy:**

Affirmed.

**Final Justification:**

After the rebuttal comments, results shown and clarifications provided, I vote for accept.

**Key Questions For Authors:**

- Can you create a side-by-side memory complexity analysis between your's and other approaches and provide explanations on the factors contributing to the specific memory utilizations observed for each?
- In the communication analysis (B7) FedRTS is missing, could you add its overhead too?
- While you demonstrate performance on GPT, and the choice of evaluated architectures is in line with related work, the architecture relevance of the evaluation could be improved. Does your method apply to more recent models/architectures such as Transformer or Mamba based architectures or even more recent CNN models? E.g. MambaVision, ConvNeXt, InternImage, DeiT, EfficientViT(by Xinyu Liu)
- Keeping the linear term in the Taylor approximation is a well motivated design choice. Can you demonstrate how the performance changes if you discard that term instead of keeping it? i.e. how big is the advantage of doing this?

**Limitations:**

No. Computing the gradients for full weights may infer significant memory cost. Also, there are approaches that are still cheaper in terms of communication. Maybe a discussion on trade-offs for on device memory requirements, communication cost and performance would be nice.

**Strengths And Weaknesses:**

## Soundness
### Strengths
- The evaluation covers relevant benchmarks and approaches and exhibits strong results for the presented method.
- The use of both vision and nlp is appreciated.
- The related work is recent and relevant.

### Weaknesses
- The evaluation could benefit from more recent models as the models are about 10 years old at this point.
- I am not sure that the logic presented for the cost is completely transparent. To my understanding, in federated learning, the memory (on edge) is constrained, which motivates the use of sparse networks to lower the memory cost. While unstructured sparsity can also benefit compute efficiency, the real world benefits might be lower. Now, the presented framework can make use of this sparsification when updating the active parameters. However, when computing the growth scores the full gradients are required to compute the kronecker factors. To me that means, that you will have the full backward complexity of the dense model. While the additional computation might amortize over several rounds of parameter updates, if peak memory consumption is the issue, this will not run anymore.
- The performance implications of how the grow score is approximated is not investigated. I.e. the importance of keeping the linear term in the approximation vs. discarding it.

## Presentation
### Strengths
- The writing quality is excellent. Following the thoughts and the design rationals is quite easy. I like the layout of figures and tables.

### Weaknesses
- I would have preferred to learn about the diagonal approximation of the KFAC factors in the main content.
- "the mask structure based on heuristic criteria, as shown in Fig. 1." (L.43) The connection between what is written and the figure is not clear to me. Should it be Fig. 2?
- How/Which grow scores are transmitted between server and client could have been discussed in the main content, since it is important to your framework, having it in the supplement is unexpected.
- Typo: Fullstop missing l.218.
- l.719: is it really substituting 36 in 30?

## Significance
The results indicate clear superiority in a problem that is relevant to federated learning. However, without further analysis on the memory overhead on the edge devices, it is unclear how easy it will be to apply to real world systems.

## Originality
While FDST uses gradient magnitude, looking at the Taylor expansion, realizing the linear term does not vanish and keeping it, which is a distinct difference basically all fisher works which are indeed focused on compression of converged models, is an interesting insight for the application. However, all of 3.2. has been done in prior work (discussed in the related works), with the key difference that the linear term is deliberately kept in this work. Moreover, while not mentioned in the main content, the use of Fisher information is not completely new to federated learning, with FedFisher demonstrating its efficacy already, which could have been more transparently stated in the main content in my opinion. Hence, the insight is nice, but the presentation oversells the contribution of the paper a little bit in my opinion.

---

> ### Author Rebuttal · Authors · 2026-03-31
>
> Dear Reviewer jwVK,
> Thank you for your thoughtful evaluation and recognition of our FedFit’s soundness and clear presentation. We appreciate the opportunity to clarify several points and address the concerns.
> > **W1&Q3**: It is important to clarify whether the method applies to more recent models and architectures or even more recent CNN models.
>
> **A:** Following your suggestion, we additionally compare FedFit with the representative FDST baseline FedDST on Qwen3.5-0.8B[1]  on the ScienceQA benchmark[2]. Because Qwen3.5-0.8B is a recent Transformer-based model. The results in the following table suggest that FedFit transfers well to newer large Transformer-based architectures, and remains effective.
> |Density|0.2|0.3|0.5|
> |---|---|---|---|
> |FedAvg|62.8|62.8|62.8|
> |FedDST|39.8|52.0|60.0|
> |FedFit|44.8|55.6|63.4|
>
> [1] Qwen Team. Qwen3.5-0.8B. Hugging Face Model Card, 2026.
>
> [2] Lu et al.,  Learn to Explain: Multimodal Reasoning via Thought Chains for Science Question Answering. NeurIPS, 2022.
>
> >**Q1&W2&W9**: The logic presented for the cost is not completely transparent. Can you create a side-by-side memory complexity analysis between your's and other approaches and provide explanations on the factors contributing to the specific memory utilizations observed for each?
>
> **A**: In terms of memory complexity, FedFit can be the same as prior methods, because the activations needed for score computation are already retained for standard backpropagation and can be reused directly. The extra memory mainly comes from the hook-based implementation, not the method itself. To support this point, we also report memory usage results below.
> |Methods|Client Peak|
> |---|---|
> |FedDST|24.9 MiB|
> |FedFit|25.6 MiB|
> |Difference|+0.7 MiB|
> >**W3&Q4**: Can you demonstrate how the performance changes if you discard that term instead of keeping it? How big is the advantage of doing this?
>
> **A**: We conduct an ablation study on ResNet18 and CIFAR-10, comparing the version that keeps the linear term with the version that discards it. show that keeping the linear term consistently improves performance across all tested densities, with an average gain of about $+2.97$ points over the three settings.
> |Density|0.1|0.2|0.3|
> |---|---|---|---|
> |FedFit keeping the linear term|65.51±1.25|74.43±1.07|76.43±1.72|
> |FedFit discarding the linear term|59.97±1.95|71.71±2.92|75.78±2.45|
>
> >**W4**: I would have preferred to learn about the diagonal approximation of the KFAC factors in the main content.
>
> **A**: We will be sure to move it from the appendix B.2. to the main content.
>
> >**W5**: “the mask structure based on heuristic criteria, as shown in Fig. 1.” (L.43) should it be Fig. 2?
>
> **A**: We appreciate your thoroughness in reading our paper. This is indeed a typo, and the correct figure is Fig.2. This will be corrected in the final manuscript.
> >**W6**: How/which grow scores are transmitted between server and client could have been discussed in the main content.
>
> **A:** We will be sure to move them from appendix C.1 to the main content
> > **W7**: Typo: Fullstop missing l.218.
>
> **A**: Thanks for pointing it out, we will revise it in the final version.
> >**W8**: L.719: is it really substituting 36 in 30?
>
> **A**: We thank you for your meticulous analysis! What we meant to say is substituting Eq. 36 into Eq. 30 after applying Eq. 31 will yield the results we stated. We will rectify this.
> > **W10**: The use of Fisher information is not completely new to federated learning. The presentation may oversell the contribution of the paper a little bit.
>
> **A**: Thank you for this helpful comment. In the revised version, we will reduce the emphasis on FIM as a standalone contribution and instead highlight our main insight, namely, that FedFit is a theoretically grounded FDST framework for online prune-and-grow adaptation under non-convergent and heterogeneous federated settings. Concretely, we will revise the second contribution in the Introduction accordingly.
> > **Q2**: In the communication analysis (B7), FedRTS is missing; its overhead should be added as well.
>
> **A**: We appreciate your rigorous reading. For FedRTS, the communication cost is $2O_s$ in the inner loop and $2O_s+0.5O_{\xi}$ in the outer loop, since it uploads only the indices of the top gradients. Under $T_{\max}=500$, $T_{\mathrm{end}}=300$, and $\Delta_T=10$, this gives a total communication cost of $1000O_s+15O_{\xi}.$

---

> > ### Author Rebuttal · Reviewer_jwVK · 2026-04-02
> >
> > Thank you for addressing my questions. In particular, the separation of linear term impact is greatly appreciated and - in my opinion - significantly strengthens the evaluation. Moreover, the results on Qwen inspire confidence in terms of generalizability.
> >
> > However, **I decided to maintain my score** because two questions remain about the memory footprint analysis.
> > 1. In your answer to W2&Q3 of 8k1S about energy consumption, you state that you have measured energy consumption but then just report memory. Moreover, given the small size of the models used, an RTX 3080 can hardly be considered a constrained device. Am I missing something there?
> >
> > 2. FedDST does not seem to perform very good in prior benchmarks vision benchmarks (Tab.1). Why was it chosen for the memory analysis over something more competitive like FedRTS? Where is your method positioned amongst the other ones evaluated in your main table?
> >
> > Upon clarification on these questions **I am prepared to increase my score.**
> >
> > After the additional answers below **I increase my score.**

---

> > > ### Author Response · Authors · 2026-04-02
> > >
> > > Dear Reviewer jwVK,
> > > Thank you for taking the time to read our rebuttals. We are glad to have addressed some of your previous questions, and we appreciate the opportunity to clarify your further concerns.
> > >
> > > > **Q1.1**: Answer to W2&Q3 of 8k1S is about energy consumption, but memory was reported.
> > >
> > > **A**: Thank you for the careful reading. Our intention was to report peak memory usage, not energy consumption, however there was a typo and energy was listed instead. We also estimated energy consumption, and the trend was consistent with memory. Because these were indirect estimates rather than direct empirical evidence, we chose to not include them in our previous rebuttal. For completeness, we provide an analytical estimate below based on our RTX 3080 Laptop GPU.
> > >
> > > For one end-to-end training round on a single client, FedDST takes $5.16 s$, while FedFit takes $5.32 s$, yielding an additional runtime of
> > > $\Delta t = 5.32 - 5.16 = 0.16 s$. We measure $115 W$ as the average edge-side power, and the extra energy per round is therefore estimated as $E_{\text{extra,round}} \approx 115 \times 0.16 = 18.4 J$. Under the default setting, training runs for $T_{\max}=500$ rounds, giving a total extra edge-side energy overhead of $E_{\text{extra,total}} \approx 500 \times 18.4 = 9200 J \approx 2.56 Wh.$
> > >
> > >
> > > > **Q1.2**: Given the small size of the models used, an RTX 3080 can hardly be considered a constrained device.
> > >
> > > **A**: As for why the RTX 3080 was used, we used GPU over CPU due to time constraints of the rebuttal period. We chose RTX 3080 as it was the weakest laptop we had ready to run our experiments. Also the device should have little effect on the memory comparison, since all methods are evaluated under the same implementation setup.
> > >
> > > We further evaluated Qwen3.5-0.8B at 0.5 sparsity on the ScienceQA benchmark using the same RTX 3080 Laptop. Since an RTX 3080 Laptop would struggle with training on Qwen3.5-0.8B without sparsity, we regard the laptop as a resource-constrained edge device. For consistency, we used our current STen-based GPU implementation to profile memory across all methods. The results show that FedFit does not introduce extra memory overhead.
> > >
> > > |Methods|Client Peak Memory|
> > > |---|---|
> > > |FedDST|6919.1 MiB|
> > > |FedFit|6924.6 MiB|
> > >
> > > > **Q2**: FedDST does not seem to perform very good in prior benchmarks vision benchmarks (Tab.1). Why was it chosen for the memory analysis? Where is FedFit positioned amongst the other ones evaluated in the main table?
> > >
> > > **A**: Our method is developed directly from FedDST, so we chose FedDST as the most appropriate baseline for the memory analysis. Comparing FedFit against FedDST allows us to isolate the overhead we introduced, without other differences in optimization pipeline or algorithmic design confounding our analysis.
> > > We also conducted experiments against stronger baselines, and the results show that our method remains competitive.
> > > |Methods|Client Peak Memory|
> > > |---|---|
> > > |FedDST|24.9 MiB|
> > > |FedTiny|24.9 MiB|
> > > |FedMef|24.2 MiB|
> > > |FedSGC|30.9 MiB|
> > > |FedRTS|24.9 MiB|
> > > |FedFit|25.6 MiB|

---

### Official Review · Reviewer_8k1S · 2026-02-28

**Soundness:** 3
**Presentation:** 3
**Significance:** 3
**Originality:** 3
**Overall Recommendation:** 4
**Confidence:** 4

**Summary:**

This paper addresses the critical communication and memory bottlenecks in cross-device Federated Learning (FL) by proposing FedFit, a Federated Dynamic Sparse Training (FDST) framework. Traditional FDST methods rely on magnitude-based heuristics, which the authors argue are ill-suited to non-convergent, heterogeneous (non-IID) environments typical of FL. FedFit replaces these heuristics with an optimization-centric approach using a second-order approximation of the loss landscape via the Fisher Information Matrix (FIM). This enables more precise parameter importance scoring for pruning and growing, allowing the discovery of efficient subnetworks directly during the FL process.

**Compliance With Llm Reviewing Policy:**

Affirmed.

**Key Questions For Authors:**

1. How does the number of local epochs ($E$) affect the bias of the Fisher Information estimate? Does more local training lead to "over-specialized" masks that hurt global aggregation?
2. Given that the current evaluation uses basic CNNs, how does the K-FAC Fisher approximation perform with the attention mechanisms in larger Transformer-based models?
3. Does the communication cost analysis explicitly account for the bit-overhead of transmitting indices for the dynamic sparse masks, especially in extremely high-sparsity regimes?

**Limitations:**

Please refer to the weaknesses and questions.

**Strengths And Weaknesses:**

Strengths:
1. The transition from heuristic-based structural adjustment to a formal second-order optimization perspective is well-motivated and provides a more principled foundation for FDST in decentralized environments.
2. The method achieves state-of-the-art results across various benchmarks (e.g., ResNet-18 on SVHN), often outperforming existing methods by significant margins (up to 7.04% in accuracy) while achieving 80% communication savings compared to dense training.

Weaknesses:
1. The experimental evaluation is primarily focused on relatively simple CNN architectures (e.g., ResNet-18, VGG-19, ShuffleNetV2) and medium-scale datasets (e.g., CIFAR-10/100, TinyImageNet). A more comprehensive study of modern, large-scale architectures such as ViT and LLMs on large-scale benchmarks (e.g., ImageNet-1k) would better demonstrate the framework's scalability in current AI landscapes.
2. While the paper claims additional computation is linear ($\mathcal{O}(n)$), calculating Fisher Information (even via K-FAC) involves extra gradient and activation statistics. The paper lacks a detailed analysis of energy consumption or peak memory usage on actual resource-constrained edge hardware.
3. The impact of the adjustment interval ($\Delta R$) and the growth ratio on the stability of the Fisher scores is not fully explored, leaving questions about how much tuning is required for different levels of data heterogeneity.

---

> ### Author Rebuttal · Authors · 2026-03-31
>
> Dear Reviewer 8k1S,
> Thank you for your thoughtful review and positive assessment of our work. We appreciate your valuable comments. Following your suggestions, we have added further experiments and clarifications to strengthen the paper.
> > **W1&Q2**: A more comprehensive study of modern, large-scale architectures on large-scale benchmarks would better demonstrate the framework’s scalability. It is also unclear about the performance in larger Transformer-based models.
>
> **A**: Our main experiments focus on CNN backbones and medium-scale benchmarks because federated pruning is primarily motivated by resource-constrained cross-device federated learning. In this setting, lightweight architectures remain highly relevant and therefore provide a practical and representative testbed for our method.
> Following your suggestion, we compare FedFit with the representative FDST baseline FedDST on Qwen3.5-0.8B[1] using the ScienceQA[2] benchmark. The results are shown below.
>
> |Density|0.2|0.3|0.5|
> |---|---|---|---|
> |FedAvg|62.8|62.8|62.8|
> |FedDST|39.8|52.0|60.0|
> |FedFit|44.8|55.6|63.4|
>
> These suggest FedFit also works well on larger Transformer-based models, and remains effective for attention-based architectures.
>
> [1] Qwen Team. Qwen3.5-0.8B. Hugging Face Model Card, 2026.
>
> [2] Lu et al., Learn to Explain: Multimodal Reasoning via Thought Chains for Science Question Answering. NeurIPS, 2022.
>
> > **W2&Q3**: Paper lacks a detailed analysis of energy consumption or peak memory usage on actual resource-constrained edge hardware. Does the communication cost analysis explicitly account for the bit-overhead of transmitting indices for the dynamic sparse masks, especially in extremely high-sparsity regimes?
>
> **A**: To address this concern, we conducted energy consumption tests on a resource constrained device. Below are our results run on a laptop running an RTX 3080 GPU. We will be sure to include this in the final version.
>
> |Methods|Client Peak|
> |---|---|
> |FedDST|24.9 MiB|
> |FedFit|25.6 MiB|
> |Difference|+0.7 MiB|
>
> Regarding communication cost, yes, our analysis already includes the overhead of sparse indices via sparse compression schemes. In particular, for density $\rho \in[0.1,0.3]$, we use COO storage, whose size is $s = m\lceil \log_2 n \rceil + mb,$ and for higher-sparsity regimes $\rho \in[0,0.1)$, we use CSR/CSC storage, whose size is $s = m\lceil \log_2 n_c \rceil + n_r\lceil \log_2 m \rceil + mb.$ Therefore, the communication cost in our paper already includes index transmission overhead, which is especially important in extremely high-sparsity regimes.
>
> > **W3&Q1**: The impact of the adjustment interval and the growth ratio on the stability of the Fisher scores is not fully explored. How does the number of local epochs affect the bias of the Fisher Information estimate? Does more local training lead to over-specialized masks that hurt global aggregation?
>
> **A**: Following the suggestion, we further study the impact of the growth ratio, the adjustment interval $\Delta_T$, and the number of local epochs on ResNet18 and CIFAR-10. The results are shown in the table below.
> | Growth Ratio | 0.1 | 0.2 | 0.3 |
> |---|---|---|---|
> | Density 0.1 | 65.60±2.36 | 65.51±1.25 | 65.50±3.36 |
> | Density 0.2 | 73.75±2.34 | 74.43±1.07 | 74.10±1.05 |
> | Density 0.3 | 75.79±2.15 | 76.43±1.72 | 75.12±2.75 |
>
> | Delta_T | 3 | 5 | 10 | 20 |
> |---|---|---|---|---|
> | Density 0.1 | 65.48±2.73 | 63.56±1.37 | 65.51±1.25 | 66.89±2.61 |
> | Density 0.2 | 74.26±1.13 | 74.40±1.93 | 74.43±1.07 | 74.66±2.94 |
> | Density 0.3 | 76.33±1.76 | 75.80±1.12 | 76.43±1.72 | 75.75±2.29 |
>
> | Local Epochs | 5 | 10 | 20 |
> |---|---|---|---|
> | Density 0.1 | 65.51±1.25 | 64.11±2.70 | 62.90±2.00 |
> | Density 0.2 | 74.43±1.07 | 73.20±4.42 | 74.65±2.10 |
> | Density 0.3 | 76.43±1.72 | 76.19±2.84 | 75.59±1.61 |
>
> The results show that FedFit is fairly robust to the growth ratio, with the default value $0.2$ giving the best or near-best performance across most densities.
> For the adjustment interval, the performance gap remains modest, which indicates that FedFit is reasonably robust to the choice of $\Delta_T = 10$.
> The results of the experiment on local epochs also suggest that excessive local training increases client drift and makes local importance estimates more client-specific. We therefore use $E=5$ as a more stable default setting and will include these studies in the appendix for the final version.
> Local epochs experiments show that indeed more local training can lead to over-specialized masks, degrading the performance of the model.

---

> > ### Author Rebuttal · Reviewer_8k1S · 2026-04-03
> >
> > Thank you to the authors for the detailed rebuttal and the additional experiments. The responses meaningfully address some of my original questions. In particular, the clarification regarding communication cost directly answers my question about the index transmission overhead of dynamic sparse masks. The additional ablations on the growth ratio, adjustment interval, and local epochs are also useful and partially address my concern about the stability of the Fisher-based importance estimates.
> >
> > That said, my concerns are only partially resolved overall. The new Transformer-based result on Qwen3.5-0.8B is a helpful signal that the approach may extend beyond CNNs, but it is still limited to a single model/benchmark pair and therefore does not fully establish scalability to modern large-scale architectures. Likewise, the new hardware evidence does not yet fully address my concern about practical deployment on genuinely resource-constrained edge devices: the reported experiment is run on a laptop with an RTX 3080 GPU, and the rebuttal still does not clearly report actual energy measurements on representative edge hardware.
> >
> > Overall, the rebuttal improves the paper and increases my confidence in some technical points, but the main concerns about real resource-constrained evaluation and broader scalability remain only partially addressed. I therefore lean toward keeping my current score unchanged. For the final version, it would be helpful to discuss more clearly the scope of the Qwen-based results and the limitations of the current hardware evaluation.

---

> > > ### Author Response · Authors · 2026-04-03
> > >
> > > Dear Reviewer 8k1S, Thank you for your time and thoughtful review. We fully respect your decision to maintain the score and are happy to have answered some of your previous questions. We also hope to address your remaining concerns.
> > >
> > > > **Q1**:  The new Transformer-based result is still limited to a single model/benchmark pair and therefore does not fully establish scalability to modern large-scale architectures.
> > >
> > > **A**: We agree that the added rebuttal results are still limited. Our goal here is to show that the method is not only restricted to CNNs, but also performs well on larger Transformer-based models. We will make these limitations explicit in the paper and leave broader scalability studies and direct energy measurements on representative edge hardware for future work.
> > >
> > > > **Q2**:  About practical deployment on genuinely resource-constrained edge devices, the rebuttal does not clearly report actual energy measurements on representative edge hardware.
> > >
> > > **A**: Thank you for bringing this concern to light. We intended to report peak memory usage rather than energy consumption, but a typo led us to list energy instead. We also estimated energy consumption, and its trend was consistent with that of memory usage. Because these estimates were indirect, we did not include them in our previous rebuttal. Below, we provide an analytical estimate based on our RTX 3080 Laptop GPU.
> > >
> > > For one end-to-end training round on a single client, FedDST takes 5.16 s, while FedFit takes 5.32 s, yielding an additional runtime of
> > > $\Delta t = 5.32 - 5.16 = 0.16 \text{s}$. We measure 115 W as the average edge-side power, and the extra energy per round is therefore estimated as $E_{\text{extra,round}} \approx 115 \times 0.16 = 18.4 \text{J}.$ Under the default setting, training runs for $T_{\max}=500$ rounds, giving a total extra edge-side energy overhead of $E_{\text{extra,total}} \approx 500 \times 18.4 = 9200 \text{J} \approx 2.56 \text{Wh}.$

---

### Official Review · Reviewer_k7RV · 2026-03-13

**Soundness:** 2
**Presentation:** 2
**Significance:** 3
**Originality:** 3
**Overall Recommendation:** 4
**Confidence:** 5

**Summary:**

The paper presents a dynamic sparse training approach based on Fisher Information-based (prune/grow) scoring for federated learning scenarios. The proposed method is evaluated across multiple tasks, datasets, and neural network architectures.

**Compliance With Llm Reviewing Policy:**

Affirmed.

**Final Justification:**

Most of my concerns have been addressed and the rating has been adjusted accordingly.

**Key Questions For Authors:**

None.

**Limitations:**

yes

**Strengths And Weaknesses:**

The utilization of the Fisher Information Matrix (FIM) in neural network pruning, as well as pruning within the federated learning framework, is not new. Although the appendix includes a related work section, the authors should clarify how the proposed approach differs from existing methods and why it is particularly important for dynamic sparse training.

Given a target density /rho, clients call PRUNEBYSCORE and GROWBYSCORE. It is noted that the pruning and recovery of parameters will eventually achieve the target density. The pruning budget is denoted by m_l; however, the growth budget is also set to m_l. Considering that participating clients receive a copy of the weights and masks from the server and use them without resetting the mask (line 988), it is unclear how many parameters are actually deactivated and subsequently reactivated during training.

The proposed approach assumes the same pruning budget for every layer. However, prior research has shown that assigning different target sparsity levels to different components of a neural network can improve performance. The authors may want to discuss why a uniform pruning budget across layers was chosen and how this choice affects performance.

The purpose of the overhead analysis (Section 3.6) is somewhat unclear. If the intention is to demonstrate that the proposed approach does not incur significantly higher overhead than existing methods, this should be stated more explicitly. While this conclusion seems somewhat evident, a direct comparison with existing approaches would help readers better understand the strengths of the proposed method.

In the experiments on the Tiny ImageNet dataset, the target density used appears unrealistic (too low), as the resulting performance drop is significantly larger than that observed for other datasets. In practice, when dealing with more complex tasks (e.g., 20 times more classes and higher input resolution), one would typically adjust the target density accordingly. Excessive pruning may remove critical parameters for any approach. Therefore, using a higher target density would allow for a more meaningful comparison of the ability of different methods to retain the most important parameters.

The pruning experimental design performs the first pruning after 5 epochs and does not consider a warm-up phase. As the authors also note, the early stage of training is unstable and it is difficult to determine the relative importance of parameters during this phase. For the datasets and models used in this study, starting pruning at epoch 5 appears somewhat early. In what scenarios would the proposed framework require pruning to begin this early? Could pruning be delayed until the model has trained for a longer period and the parameter importance becomes more stable?

It appears that 300 iterations were used as the total training duration. How was this value selected? Although early stopping may not be typical in federated learning with pruning, it would be helpful to clarify what indicators were used to determine when to stop training at both the local and global levels, particularly to prevent excessive training.

Other comments
- Page 1, second column, line 44: Figure 1 does not appear to illustrate what is described in the sentence.
- Page 3, first column, line 154: Figure 1 again does not clearly correspond to the description in the text. If the authors are referring to a “0.5 improvement in accuracy” as an empirical result, the source of this result (whether from the authors’ experiments or from the literature) should be clarified.
- Subsection 3.1 (line 146): It is common practice that pruning does not occur in the early stage of training. Instead, pruning typically begins after a warm-up phase.

---

> ### Author Rebuttal · Authors · 2026-03-31
>
> Dear Reviewer k7RV,
> Thank you for your thoughtful review. We appreciate the opportunity to clarify several points and address the concerns.
> > **W1**: The utilization of the FIM in neural network pruning and federated learning framework is not new.
>
> **A**:  Our novelty does not come from using FIM itself, but from applying it to a different problem with a different motivation. Prior Fisher-based methods such as FedFisher [1] study post-training pruning for converged models, whereas FedFit is designed for FDST, where topology is updated online under non-IID and non-convergent conditions. In this setting, the linear term should be retained, as Reviewer jwVK also pointed out. We also provide, to our knowledge, the first systematic theoretical account of the criterion underlying FDST, and derive a unified framework for both pruning and growth, which Reviewer uuDJ also recognized.
>
> [1] Divyansh et al.,Fedfisher: Leveraging fisher information for one-shot federated learning. International Conference on Artificial Intelligence and Statistics. AISTAT, 2024.
>
> > **W2**: It is unclear how many parameters are actually deactivated and subsequently reactivated during training.
>
> **A**: Following our notation, at adjustment step $t$, the pruning & growing number for layer $l$ is  $\zeta_t\rho_l n_{l},$
> where $\rho_l$ is layer-wise density,  $n_l$ is the number for weights for $l$,  $\zeta_t=0.2\left(1+\cos\frac{t\pi}{T_{end}}\right)$, following previous work, FedRTS.
>
> > **W3**: The authors may discuss why a uniform pruning budget across layers was chosen and how this choice affects performance.
>
> **A**: Thank you for your careful reading. We would like to clarify we use ERK for layer-wise sparsity allocation. Concretely, the layer-wise sparsity is determined by the ERK rule $\rho_l \propto \frac{n_{in,l}+n_{out,l}}{n_{in,l}n_{out,l}},$ This detail can be verified in our released code at **experiments/distributed/fedfit/main\_fedfit.py** (line 94). We additionally compare ER, ERK, and Uniform allocation on ResNet18 and CIFAR-10. We will clarify this more explicitly in the revised version.
> |Density|0.1|0.2|0.3|
> |---|---|---|---|
> |Uniform|59.85±2.97|69.74±2.31|71.49±0.62|
> |EK|63.01±1.85|72.14±0.77|74.10±1.72|
> |ERK|65.51±1.25|74.43±1.07|76.43±1.72|
>
> [1] Evci et al., Rigging the lottery: Making all tickets winners.ICML 2020.
>
> > **W4**: The purpose of the overhead analysis is somewhat unclear. A direct comparison with existing approaches would strengthen the method.
>
> **A**: Our intention was to show that the proposed method does not introduce high overhead. Since activations are retained during backpropagation, they can be reused directly. To support this point, we report memory usage results below.
> |Methods|Client Peak|
> |---|---|
> |FedDST|24.9MiB|
> |FedFit|25.6MiB|
> |Difference|+0.7MiB|
> > **W5**: Paper should increase density on TinyImageNet .
>
> **A**: We added experiments on TinyImageNet with higher target densities which shows FedFit remains effective.
> |Method|ResNet18 (0.4)|ResNet18 (0.5)|ShuffleNet (0.4)|ShuffleNet (0.5)|
> |---|---|---|---|---|
> |FedAVG|8.97±0.23|8.97±0.23|5.87±0.45|5.87±0.45|
> |FedFit|8.15±0.21|8.77±0.45|5.70±0.53|5.85±0.65|
> > **W6&W11**: In what scenarios the proposed framework requires pruning to begin this early? Could pruning be delayed after a warmup?
>
> **A**: We begin prune-and-grow earlier because it gives the model more chances to explore different sparse subnetworks. Results shown in the table show that long warm-up does not reliably improve performance.
> |Warmup Epochs|0|25|50|100|
> |---|---|---|---|---|
> |Density 0.1|65.51±1.25|62.59±3.36|63.42±3.67|61.64±3.00|
> |Density 0.2|74.43±1.07|74.53±2.98|73.21±5.10|72.79±4.64|
> |Density 0.3|76.43±1.72|74.23±6.41|74.56±2.33|73.36±3.26|
> > **W7**:  How is the total training duration of 300 iterations selected?
>
> **A**: We clarify that our method uses a total budget of $T_{max}=500$ communication rounds, with structure adjustment applied during the first $T_{end}=300$ rounds. Experiments shown below suggest larger training budgets do not consistently improve performance.
> |Pruning / Total Epochs|300 / 500|500 / 700|800 / 1000|1000 / 1200|
> |---|---|---|---|---|
> |Density 0.1|65.51±1.25|64.73±4.42|65.09±3.88|64.98±3.15|
> |Density 0.2|74.43±1.07|72.14±3.14|73.64±3.56|74.36±2.64|
> |Density 0.3|76.43±1.72|77.34±1.44|77.11±3.80|76.69±1.19|
> > **W8**: Page 1,line 44, Figure 1 does not appear to illustrate what is described.
>
> **A**: Thanks for pointing it out, this should refer to figure 2. We will revise it.
> > **W9**: Page 3, line 154, Figure 1 does not clearly correspond to the text.
>
> **A**: We appreciate your feedback, this should refer to Table 1. It will be corrected for the final version.
> > **W10**: The source of 0.5 improvement in accuracy should be clarified.
>
> **A**: It comes from experiment results. We chose FedTiny as the representative SOTA method and computed avg(FedTiny) - avg(FSST) . We receive 0.53 and round to 0.5.

---

> > ### Author Rebuttal · Reviewer_k7RV · 2026-04-04
> >
> > Thank you for the responses and the additional experimental results provided within a limited timeframe. Most of my concerns have been addressed. However, the following points remain open. If these remaining issues are clarified, I would be willing to raise my rating to 4.
> >
> > On W7, my original concern was not whether extended training improves performance, but rather how FedFit determines when the global model has overfit and how that decision affects test-time performance. Demonstrating that longer training does not improve accuracy addresses a different point. Moreover, the new experiments show varying and inconsistent performance of FedFit across different T_max and T_end settings. It is therefore important to clearly explain how these parameters were chosen and whether this constitutes a fair choice for comparison with the baseline approaches.
> >
> > Additionally, in the new experiments, some baseline approaches are missing, and in the figures, not all baselines are consistently reported (e.g., Figures 4 and 6 do not include results for FedRTS). If any baseline was omitted for a specific reason, this should be explicitly stated. Omitting baselines without explanation may create a misleading impression of FedFit’s relative performance.

---

> > > ### Author Response · Authors · 2026-04-07
> > >
> > > Dear Reviewer k7RV,
> > > Thank you for taking the time to read our rebuttal and increasing the score to 4. We are happy to have addressed most of your previous questions, and hope that the following will address your remaining concerns.
> > >
> > >
> > > > **Q1.1**: How FedFit determines when the global model has overfit and how that decision affects test-time performance.
> > >
> > > **A**: Thank you for the question. FedFit uses the validation dataset during training to help ensure that the global model does not overfit.
> > >
> > > > **Q1.2**: The new experiments show varying and inconsistent performance of FedFit across different $T_{max}$ and $T_{end}$ settings. How were parameters for experiments chosen and whether this constitutes a fair choice for comparison with the baseline approaches.
> > >
> > > **A**: Thank you for the question. The choice of $T_{max}$ and $T_{end}$ follows the settings used in the prior SOTA method, FedRTS[1], rather than being tuned specifically for FedFit. We made this choice to ensure a fair comparison with the baselines under a consistent experimental setting. The new experiments in rebuttal with different $T_{max}$ and $T_{end}$ settings are included only to show that FedFit is not overly sensitive to these hyperparameters, not to optimize the method for the reported comparisons. We will make this point clearer in the revision.
> > >
> > > [1] Huang, et al. "Fedrts: Federated robust pruning via combinatorial thompson sampling." arXiv preprint arXiv:2501.19122 (2025).
> > >
> > > > **Q2**: In the new experiments, some baseline approaches are missing, and in the figures, not all baselines are consistently reported.
> > >
> > > **A**: Thank you for pointing this out. Below, we supplement W4 of our previous rebuttal by including the other baselines for reference. The results show that our method remains competitive.
> > >
> > > |Methods|Client Peak Memory|
> > > |---|---|
> > > |FedDST|24.9MiB|
> > > |FedTiny|24.9MiB|
> > > |FedMef|24.2MiB|
> > > |FedSGC|30.9MiB|
> > > |FedRTS|24.9MiB|
> > > |FedFit|25.6MiB|
> > >
> > > We also supplement W5 of our previous rebuttal with additional TinyImageNet experiments at higher target densities, including FedSGC, FedMef, and FedRTS, since these are the best performing baselines in Table 1. We also include FedDST because FedFit is built directly on top of it, making it the best reference for isolating the effect of our modification. The updated results show that FedFit remains effective under these settings.
> > >
> > > |Method|ResNet18 (0.4)|ResNet18 (0.5)|ShuffleNet (0.4)|ShuffleNet (0.5)|
> > > |---|---|---|---|---|
> > > |FedAVG|8.97±0.23|8.97±0.23|5.87±0.45|5.87±0.45|
> > > |FedDST|7.91±0.22|8.62±0.29|5.48±0.20|5.90±0.47|
> > > |FedMef|8.06±0.32|8.30±0.26|5.68±0.34|5.85±0.52|
> > > |FedSGC|8.12±0.32|8.76±0.24|5.60±0.35|5.68±0.51|
> > > |FedRTS|8.13±0.34|8.76±0.49|5.63±0.47|5.86±0.43|
> > > |FedFit|8.15±0.21|8.77±0.45|5.70±0.53|5.85±0.65|
> > >
> > > For the results in our paper. Figure 6 is intended as an ablation study. Since it focuses on analyzing the effect of our design relative to the base method, we reported FedDST (base model) and FedFit only.
> > > For Figures 3, 4, and 5, we agree that the presentation should be more consistent. We therefore conducted additional experiments and updated these figures accordingly. The updated Figures 3, 4, and 5 are shown below, and they indicate that FedFit still outperforms all baselines in most cases.
> > >
> > > https://anonymous.4open.science/r/FedFit-ED8F/Perplexity_plot.png
> > >
> > > https://anonymous.4open.science/r/FedFit-ED8F/Partition_alpha_plot.png
> > >
> > > https://anonymous.4open.science/r/FedFit-ED8F/Participation_plot.png

---

### Official Review · Reviewer_uuDJ · 2026-03-13

**Soundness:** 3
**Presentation:** 3
**Significance:** 3
**Originality:** 3
**Overall Recommendation:** 5
**Confidence:** 3

**Summary:**

This paper points out that existing FDST methods typically use weight magnitude for pruning and gradient magnitude for growth, but in federated learning settings that are non-converged, non-IID, and highly heterogeneous across clients, the assumptions underlying these heuristic rules, such as near convergence of the model and relatively stable local curvature, may no longer hold. The authors therefore argue that prune and grow rules should explicitly take second-order information into account. To this end, the paper derives pruning scores and growth scores directly from a second-order approximation of the loss, then uses the Fisher information matrix to approximate the Hessian, and further combines this with K-FAC to obtain a tractable implementation together with corresponding theoretical analysis. Finally, the method is evaluated experimentally on both computer vision and NLP tasks.

**Compliance With Llm Reviewing Policy:**

Affirmed.

**Final Justification:**

The rebuttal addresses my main concerns, and the additional experiments and theoretical clarifications further strengthen the paper; accordingly, I am increasing my confidence while keeping my overall recommendation as Accept.

**Key Questions For Authors:**

Could the authors provide some more intuitive toy examples or small-scale visual analyses to illustrate why existing heuristic prune and grow rules may fail in FDST? Although the paper offers a reasonable intuition for why magnitude-based pruning and gradient-based growth can become unreliable under non-IID and non-converged federated training conditions, this part is still developed mainly at the level of theory and narrative explanation. Adding such examples would help support the motivation more directly and make the advantages of the proposed method more convincing

**Limitations:**

Yes

**Strengths And Weaknesses:**

Strengths

1. The problem is clearly motivated and practically important.  The paper targets a real challenge in federated dynamic sparse training. Existing methods typically use weight magnitude for pruning and gradient magnitude for growth, but these heuristic rules are largely inherited from the post-training pruning setting, where the model is usually assumed to be near convergence, gradients are small, and the local curvature is relatively stable. In federated learning, however, the training process is often non-IID, highly heterogeneous, and far from convergence, so these assumptions may no longer hold. In this sense, the problem considered in the paper is well motivated and important.

2. The method is well motivated, and the theoretical derivation is relatively complete. The core idea is clear. Since the near-convergence assumptions underlying traditional prune and grow criteria may be unreliable in FDST, the paper instead derives pruning and growth scores directly from a second-order approximation of the loss. The Hessian is then approximated using Fisher information, together with a K-FAC-based tractable approximation. This derivation is coherent and also gives pruning and growth a unified treatment within the same framework.

3. The experimental coverage is relatively comprehensive. The paper includes not only computer vision tasks but also NLP tasks. Relative to prior work on federated sparse training, the experimental scope is fairly broad and generally supports the main claims about the effectiveness of the method.

Weaknesses

1. The paper lacks more direct reporting of the practical overhead. At present, the paper mainly provides theoretical analysis in terms of asymptotic complexity and additional cost, but gives less direct evidence about actual training time, memory usage, and how much extra computation is introduced on the client side. Since the method relies on Fisher approximation and K-FAC statistics, these practical costs are very important

2. The empirical explanation of why existing methods fail could be stronger. The paper provides a reasonable intuition for why previous heuristic prune and grow rules may fail in FDST, but this part still remains mostly at the level of theory and narrative explanation. It would strengthen the paper if the authors could include more direct toy examples or small-scale visual analyses showing how magnitude-based pruning or gradient-based growth deviates from the desired behavior under non-IID and non-converged conditions.

---

> ### Author Rebuttal · Authors · 2026-03-31
>
> Dear Reviewer uuDJ,
> Thank you for your time and effort in reading our paper, and for the high assessment you have given us! We really appreciate the valuable insights you have given to us, and have implemented them to help improve our paper.
> Weaknesses
> > **W1**: Lacks more direct reporting of the practical overhead. Gives less direct evidence about actual training time, memory usage, and how much extra computation is introduced on the client side.
>
> **A**: Thank you for this important insight on the practicality of our computational overhead. To measure this, we have conducted some experiments, evaluating time and memory, to back our theoretical costs. Below are our results run on a laptop running an RTX 3080 GPU.
>
> |Methods|Client Peak|
> |---|---|
> |FedDST|24.9 MiB|
> |FedFit|25.6 MiB|
> |Difference|+0.7 MiB|
>
> End-to-End Training time for one training round of one client is shown in the following table, which demonstrates the minimal overhead of FedFit.
>
> | Methods | Time |
> |---|--|
> | FedDST | 5.16 s |
> | FedFit | 5.32 s |
> |Difference|0.16 s|
>
> > **W2&Q1**:  The empirical explanation of why existing methods fail could be stronger.
> Could authors provide some more intuitive toy examples or small-scale visual analyses to illustrate why existing heuristic prune and grow rules may fail in FDST?
>
> **A**: We appreciate the thoughtfulness and attention to detail of the reviewer. We further define a robustness score $R$ to quantify retained accuracy under heterogeneity, $d = acc_{IID} - acc_{0.1}\$, where 0.1 denotes the partition alpha, $\hat{d} = \frac{d}{acc_{IID}}$, and $R = 1 - \hat{d}$. This is consistent with prior work that measures relative robustness as one minus the normalized performance drop, with higher values indicating stronger robustness[1]. The results in figure
> https://anonymous.4open.science/r/FedFit-ED8F/Robustness_plot.png
> demonstrates that existing heuristic methods fail in FDST due to lower robustness.
>
> [1] Schiappa et al., Robustness Analysis on Foundational Segmentation Models.CVPRW, 2024.

---

> > ### Author Rebuttal · Reviewer_uuDJ · 2026-04-01
> >
> > Thank you for the rebuttal. It addresses my main concerns. I am keeping my current overall recommendation as Accept.

---

> > > ### Author Response · Authors · 2026-04-03
> > >
> > > Thanks for your feedback! We're pleased that you are satisfied with the response and keep your positive score. We sincerely appreciate your suggestions, which have helped improve the clarity and readability of the paper. We will incorporate these discussion points into the final version.

---

### Decision · Program_Chairs · 2026-04-30

**Decision:**

Accept (regular)

**Comment:**

This paper proposes FedFit, a federated dynamic sparse training method that replaces standard heuristic prune/grow rules with scores derived from a second-order loss approximation. The key motivation is compelling: in federated settings, models are often non-converged, non-IID, and heterogeneous, so the assumptions underlying standard magnitude- and gradient-based heuristics are weaker than in conventional sparse training or post-training pruning.

Across the reviews, there is broad agreement that the problem is important, the paper is well motivated, and the method is technically interesting. Reviewers positively highlighted the principled treatment of pruning and growth within a unified second-order framework, the explicit retention of the linear term for the non-converged federated setting, and the breadth of experiments across both vision and NLP tasks. Several reviewers also found the rebuttal helpful in clarifying the distinction from prior Fisher-based methods, strengthening the empirical case, and addressing questions about memory overhead, robustness, and the role of the linear term.

The main concerns were about practical overhead and scalability. In particular, reviewers asked for clearer evidence about memory and runtime overhead on clients, stronger discussion of broader architectural scalability, and more transparent positioning relative to prior Fisher-based pruning work. Based on the rebuttal and discussion, I believe these issues were addressed sufficiently for acceptance. The authors provided additional measurements showing modest memory/runtime overhead relative to closely related baselines, added a Transformer-based result beyond the CNN setting, and included ablations that directly support one of the paper’s central design choices—namely, retaining the linear term. At the same time, I agree that the current evidence on large-scale architectures and truly resource-constrained hardware remains limited, and these limitations should be stated clearly in the final version.

Overall, I find this to be a solid and well-supported contribution. The paper offers a principled improvement over heuristic FDST criteria, shows meaningful empirical gains, and was positively evaluated by the reviewers after rebuttal. For the final version, I encourage the authors to more clearly position the work relative to prior Fisher-based pruning in FL, explicitly discuss the limits of the current hardware/scalability evidence, and integrate the rebuttal-added overhead and ablation results into the paper.